# ONE2SCENE: GEOMETRIC CONSISTENT EXPLORABLE 3D SCENE GENERATION FROM A SINGLE IMAGE

**Pengfei Wang**[*], **Liyi Chen**[*], **Zhiyuan Ma, Yanjun Guo, Guowen Zhang, Lei Zhang**[†]
The Hong Kong Polytechnic University
`pengfei.wang@connect.polyu.hk, cslzhang@comp.polyu.edu.hk`
[*]Equal contribution. [†]Corresponding author.
Project page: `https://one2scene5406.github.io/`

## ABSTRACT

Generating explorable 3D scenes from a single image is a highly challenging problem in 3D vision. Existing methods struggle to support free exploration, often producing severe geometric distortions and noisy artifacts when the viewpoint moves far from the original perspective. We introduce **One2Scene**, an effective framework that decomposes this ill-posed problem into three tractable sub-tasks to enable immersive explorable scene generation. We first use a panorama generator to produce anchor views from a single input image as initialization. Then, we lift these 2D anchors into an explicit 3D geometric scaffold via a generalizable, feed-forward Gaussian Splatting network. Instead of treating the panorama as a single image for reconstruction, we project it into multiple sparse anchor views and reformulate the reconstruction task as multi-view stereo matching, which allows us to leverage robust geometric priors learned from large-scale multi-view datasets. A bidirectional feature fusion module is used to enforce cross-view consistency, yielding an efficient and geometrically reliable scaffold. Finally, the scaffold serves as a strong prior for a novel view generator to produce photorealistic and geometrically accurate views at arbitrary cameras. By explicitly conditioning on a 3D-consistent scaffold to perform reconstruction, One2Scene works stably under large camera motions, supporting immersive scene exploration. Extensive experiments show that One2Scene substantially outperforms state-of-the-art methods in panorama depth estimation, feed-forward 360° reconstruction, and explorable 3D scene generation.

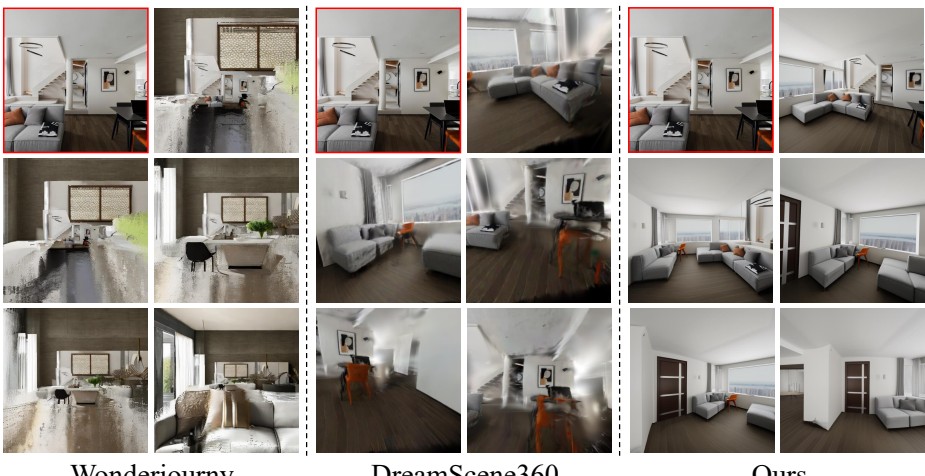

Figure 1: **Comparison on large-viewpoint novel view synthesis.** Existing methods such as Wonderjourny (Yu et al., 2023) and Dreamscene360 (Zhou et al., 2024) exhibit clear geometric distortions and artifacts, while our method generates photorealistic and geometrically accurate novel views. The input image is highlighted by a red bounding box. The other images represent the novel views.

# 1 INTRODUCTION

The increasing demand for high-quality 3D content is reshaping the landscape of video games, visual effects, mixed reality, and 3D scene understanding (Wang et al., 2024a), making 3D generation a highly active research topicm (Valevski et al., 2024; Adamkiewicz et al., 2022; Martin-Brualla et al., 2021; Ye et al., 2024b; Chen et al., 2025a). Reconstruction-based methods like Neural Radiance Fields (NeRF) (Mildenhall et al., 2020) and Gaussian Splatting (GS) (Kerbl et al., 2023) have achieved remarkable results, but they typically require hundreds or even thousands of input images. Although sparse-view reconstruction approaches alleviate this requirement (Wang et al., 2023; Yang et al., 2023; Yu et al., 2024a; Charatan et al., 2024; Liu et al., 2024c;b; Wu et al., 2024a; Szymanowicz et al., 2024b), these methods struggle with large viewpoint extrapolation and fail to generalize to unseen regions. In stark contrast, generative view synthesis (Liu et al., 2023; Sargent et al., 2024; Liu et al., 2024a; Yu et al., 2024b; Li et al., 2025b) is emerging as a significant advancement in 3D content creation, as it can generate plausible content in unobserved regions (Shi et al., 2024; Zhou et al., 2025; Szymanowicz et al., 2025).

Although object-level 3D generation (Liu et al., 2023; Sargent et al., 2024; Ye et al., 2024b) has achieved rapid progress, generating an explorable 3D scene from a single image remains a significant challenge. One of the key challenges is how to maintain 3D geometric consistency and visual quality under large viewpoint changes and long-term generation. Some methods leverage pre-trained video generation models (Brooks et al., 2024; Xing et al., 2024; Hong et al., 2022; Yang et al., 2024) to create 3D-aware sequences (Liu et al., 2024a;d; Yu et al., 2024b; Chen et al., 2024; Sun et al., 2024; Liang et al., 2024), but they often suffer from geometric inconsistency and loop-closure consistency. Panorama-based pipelines such as Dreamscene360 (Zhou et al., 2024) and DreamCube (Huang et al., 2025) attempt to convert panoramas into 3D scenes, but their ability to support broader exploration is very limited, as shown in Figure 1 (a). Although navigation and inpainting-based methods (Chung et al., 2023; Yu et al., 2023; Höllein et al., 2023) enable the generation of more expansive scenes, their iterative nature often causes global semantic drift. Furthermore, cumulative errors often result in stretched or distorted geometry, as shown in Figure 1 (b). These limitations highlight the need for a new approach that can produce geometrically accurate and photorealistic scenes from a single image while supporting broad exploration.

To achieve the goal mentioned above, in this paper we introduce **One2Scene**, a novel framework that systematically decomposes explorable 3D scene generation into three distinct, yet more manageable subtasks. First, to overcome the profound information deficit of a single image, we generate a set of anchor views for global coverage using a panoramic cubemap representation. Note that these anchor views alone are insufficient to create a truly explorable scene, as shown in Figure 1 (a). Full exploration requires synthesizing high-quality novel views from arbitrary viewpoints, while how to ensure 3D consistency presents a significant hurdle. To this end, we introduce a powerful and efficient prior that encodes both geometry and appearance to stably constrain the generative process. Specifically, we reformulate the problem of monocular panoramic depth estimation as a multi-view stereo matching problem across extremely sparse anchor views, and lift the 2D anchor views into an explicit 3D geometric scaffold using a feed-forward 3D GS model. Such a design not only ensures the high efficiency of our feed-forward model but also critically enables us to leverage robust geometric priors learned from large-scale multi-view datasets. To further enforce geometric consistency across anchor-view boundaries, we introduce a bidirectional fusion module. As a result, our feed-forward model can reconstruct a geometrically accurate, high-quality 3D scaffold in 0.5 seconds.

The constructed explicit geometric scaffold provides strong priors for both geometry and appearance to guide the final novel view synthesis. To effectively utilize this scaffold, we introduce a novel Dual-LoRA training strategy. Unlike common refinement models that use channel-wise conditional injection (Wu et al., 2025), our strategy effectively fuses information from the high-quality input view with the coarse yet geometrically-rich views rendered from our scaffold. These combined conditions then guide the generation process at arbitrary camera views via a global 3D-aware attention mechanism. Our experiments demonstrate that this design significantly enhances the model's ability to leverage the priors provided. By grounding the generation process in a consistent 3D representation, the final results of our One2Scene model are not only photorealistic but also exhibit superior multi-view consistency, as demonstrated in Figure 1 (c).

Our contributions can be summarized as follows. First, we introduce a powerful feed-forward 3D GS model with a bidirectional fusion module to construct a high-quality 3D scaffold by reformulating

the monocular panoramic depth estimation into a multi-view stereo problem. Second, we present a scaffold-guided synthesis method to utilize explicit geometric and appearance priors from any target view, which robustly grounds the final rendering and resolves the geometric ambiguities inherent in single-image generation. Finally, we demonstrate that our proposed One2Scene sets a new state-of-the-art on explorable 3D scene generation, achieving superior photorealism and geometric accuracy, particularly under significant viewpoint shifts.

## 2 RELATED WORK

**3D Scene Reconstruction.** While differentiable rendering techniques such as NeRF (Mildenhall et al., 2020) and 3DGS (Kerbl et al., 2023) have achieved remarkable results, they are primarily tailored for per-scene optimization requiring dense input views, a constraint that hinders their deployment in real-world scenarios. In response, the research community has introduced various methods for sparse-view reconstruction (Wang et al., 2023; Yang et al., 2023; Yu et al., 2024a; Charatan et al., 2024; Liu et al., 2024c;b; Wu et al., 2024a; Szymanowicz et al., 2024b). Concurrently, generalizable feed-forward models (Charatan et al., 2024; Chen et al., 2025b; Szymanowicz et al., 2024b;a; Wewer et al., 2024; Xu et al., 2025; Ye et al., 2024a; Hong et al., 2024; Tang et al., 2024) have garnered significant attention for their ability to directly infer 3D representations from sparse inputs without per-instance optimization. Despite these advancements, these methods share a critical bottleneck: a lack of extrapolation capability, resulting in an inability to plausibly render unobserved regions.

**Video Diffusion-based 3D Scene Generation.** Recent video generation models (Brooks et al., 2024; Xing et al., 2024; Hong et al., 2022; Yang et al., 2024; Wan et al., 2025) have shown great potential to generate 3D-aware sequences. These models can naturally serve as 3D scene generators when camera poses are controllable (Guo et al., 2024; Wang et al., 2024c; Melas-Kyriazi et al., 2024; Voleti et al., 2024; Liang et al., 2024). To enhance 3D consistency, contemporary approaches like ReconX (Liu et al., 2024a), ViewCrafter (Yu et al., 2024b), and VMem (Li et al., 2025a) integrate explicit 3D geometric priors into their frameworks, leveraging robust reconstruction backbones such as DUSt3R (Wang et al., 2024b) and CUT3R (Wang et al., 2025b). However, despite these advancements, such methods remain limited in large-scale explorable scene generation, where the accumulation of reconstruction errors leads to geometric collapse.

**Image Diffusion-based 3D Scene Generation.** Several innovative investigations (Liu et al., 2023; Wu et al., 2024b; Sargent et al., 2024; Höllein et al., 2024; Seo et al., 2024; Shi et al., 2024; Wang & Shi, 2023; Shi et al., 2023; Liu et al., 2024e) have incorporated camera pose information into pre-trained T2I models to generate novel views. Within this category, two key strategies have emerged for generating explorable scenes from a single image. The first strategy employs **pose-conditioned view synthesis**. Methods such as SEVA (Zhou et al., 2025) and CAT3D (Gao et al., 2024) leverage camera pose information to guide the generation of novel views, demonstrating impressive scene-level results. However, when applied to single-image inputs over extended camera trajectories, these methods struggle to maintain long-range geometric consistency and visual coherence, often resulting in accumulated errors and semantic drift that compromise global scene structure. The second strategy relies on **iterative navigation and inpainting**(Pu et al., 2024; Chung et al., 2023; Yu et al., 2023; Höllein et al., 2023). One notable example, Pano2Room(Pu et al., 2024), builds the scene sequentially by navigating through space and inpainting unseen areas. Although it can produce plausible indoor results, this iterative framework is inherently prone to accumulating geometric and appearance errors over time, compromising global scene consistency. A second limitation is its design, which incorporates strong indoor priors that restrict its generalization to outdoor scenes and diverse visual styles.

In contrast to these sequential approaches, our One2Scene framework introduces a novel scaffold-guided paradigm. It decomposes the ill-posed single-image-to-scene problem into more manageable subtasks, achieving superior geometric fidelity and photorealistic quality. By first generating a globally consistent 3D scaffold in a single, feed-forward pass, our One2Scene method establishes a robust geometric and semantic foundation for the entire scene. This holistic global prior directly counteracts the error accumulation inherent in sequential methods like pose-conditioned synthesis and iterative inpainting. Consequently, our approach is not only more geometrically consistent but also significantly more general than specialized methods like Pano2Room, demonstrating superior performance across both indoor and outdoor environments.

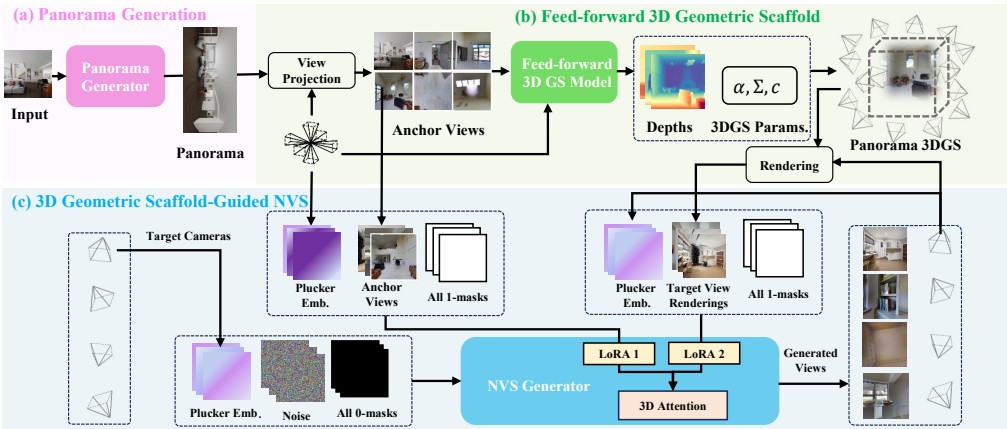

Figure 2: **Overview of One2Scene.** Our method consists of three stages: (a) an anchor view generation stage to establish an initial 360-degree representation, (b) a feed-forward 3D Gaussian Splatting stage to construct an explicit 3D geometric scaffold, and (c) a synthesis stage that leverages the scaffold information to produce high-quality novel views. The pipeline enables geometrically consistent and photorealistic novel view synthesis from a single input image.

# 3 METHODOLOGY

This section details our One2Scene framework, which can generate an explorable 3D scene from a single image by decomposing this ill-posed problem into a sequence of manageable sub-tasks, as illustrated in Figure 2. First, to overcome the severe information deficit, we generate a panorama to cover the global scene. Second, we obtain a set of anchor views from the panorama and introduce a feed-forward 3D GS model to lift these 2D anchor views into an explicit 3D geometric scaffold. Finally, with the strong geometric and appearance priors provided by the 3D scaffold, a synthesis network is used to generate photorealistic and consistent novel views from arbitrary camera poses.

## 3.1 PANORAMA GENERATION

Generating explorable 3D scenes from a single image is a highly challenging problem, often resulting in pronounced semantic drift and geometric inconsistency across long-range novel views. To address this challenge, we adopt a progressive approach that first expands visual information content and subsequently establishes a robust geometric foundation. We employ a specialized image-to-panorama generation model to transform the limited input view into a 360° panoramic representation. This representational choice is motivated by two primary considerations. First, the comprehensive field of view provides more visual cues that facilitate subsequent globally consistent scene generation. Second, compared to direct arbitrary novel view synthesis, panoramic image generation with a single image as input is a more well-posed computational task. In particular, we employ Hunyuan-Pano-DiT (Wang et al., 2025c), which demonstrates exceptional generalization capabilities acquired through training on extensive large-scale datasets, to generate the panoramic image.

## 3.2 FEED-FORWARD 3D GEOMETRIC SCAFFOLD

Although the panorama generated from the initial stage provides global coverage, it remains a 2D representation estimated from a single viewpoint and lacks explicit 3D information. Maintaining geometric consistency when synthesizing with large viewpoint changes and long sequences remains a fundamental challenge in explorable scene generation. To this end, we introduce a novel feed-forward 3D GS model to predict a set of 3D Gaussian parameters $(\boldsymbol{\mu}_i, \alpha_i, \boldsymbol{\Sigma}_i, \boldsymbol{c}_i), i = 1^{H \times W \times N}$ for each pixel in the generated panorama. This process provides the scene with explicit 3D information, thereby ensuring global geometric consistency.

**Anchor View Projection.** Accurate depth estimation is the cornerstone of this model, as inaccurate depth can introduce severe rendering artifacts. Although significant progress has been made in depth estimation from a single panoramic image (Ai et al., 2023; Wang & Liu, 2024; Pintore et al.,

2023), this task remains highly challenging. A key difficulty lies in the lack of large-scale datasets comparable to those available for perspective images, limiting the generalization ability of panoramic depth estimators. To achieve robust depth estimation, we propose to reformulate the problem of monocular panoramic depth estimation as a multi-view stereo matching problem. Specifically, we first project the 360° panorama into a set of six perspective cubemap views, which serve as the input anchor views for our model. This strategy allows us to leverage powerful geometric priors learned from large-scale multi-view datasets. We choose to use cubemaps because they provide the most compact perspective representation of the panoramic scene, ensuring high efficiency. To facilitate correspondence matching across views, we expand each cubemap's Field of View (FoV) to 95°, creating a 2.5° overlap at adjacent view boundaries. For further details, please refer to **??**.

**Bidirectional Fusion Module**. Although a 2.5-degree overlap is established between adjacent anchor views, the correspondence remains extremely sparse. Existing multi-view stereo models like VGGT (Wang et al., 2025a), which rely on substantial inter-view overlap, suffer from significant performance degradation in such scenarios. To address this limitation, we propose novel architectural modifications to VGGT to explicitly enforce cross-view consistency and improve the robustness of depth estimation. Specifically, we integrate a bidirectional fusion mechanism into the pre-trained DPT head of VGGT to promote cross-view depth consistency. This mechanism establishes geometric correspondence across views while preserving view-specific details.

To effectively handle overlapped regions, we introduce a Cube-to-Equirectangular (C2E) transformation module that projects the dense feature maps $\mathbf{F}_i$ from the six anchor views into a unified equirectangular latent. Subsequently, these equirectangular features are fused using a convolutional layer $\mathbf{H}_c$. Then, the fused features $\mathbf{F}_e$ are transformed back to the cubic space via an Equirectangular-to-Cube (E2C) module and merged with the original anchor view features through a residual connection. The finally updated feature for each view, $\mathbf{F}'_i$, is computed as follows:

$$\mathbf{F}_e = \mathbf{H}_c(\text{C2E}(\{\mathbf{F}_i\}_{i=1}^6)), \quad \mathbf{F}'_i = \mathbf{F}_i + \text{E2C}(\mathbf{F}_e). \tag{1}$$

This bidirectional transformation and fusion mechanism aligns features in overlapped regions to achieve geometric consistency via C2E/E2C transformations, while using residual connections to maintain view-specific details simultaneously. For further details, please refer to **??**.

**Gaussian Parameter Prediction Heads**. For each pixel, the Gaussian center $\boldsymbol{\mu}$ is computed by unprojecting the predicted depth into 3D space using the camera intrinsics: $\boldsymbol{\mu} = \mathbf{K}^{-1}\boldsymbol{u}d + \Delta$, where $\mathbf{K}$ denotes the camera intrinsic matrix, $\boldsymbol{u} = (u_x, u_y, 1)$ represents the pixel coordinates, and $\Delta \in \mathbb{R}^3$ indicates the predicted positional offset. To predict the remaining Gaussian parameters (opacity, covariance, and color), we employ an additional prediction head based on the DPT architecture. Following NoPosplat (Ye et al., 2024a), this prediction head takes both VGGT features and the RGB image as inputs. The direct pathway from RGB images complements VGGT's high-level semantic-focused features by preserving essential fine textural details.

**Training**. The feed-forward 3DGS model is trained using a composite loss function, which includes a rendering loss and a depth loss. The rendering loss is a combination of the Mean Squared Error (MSE) and the LPIPS perceptual loss (Johnson et al., 2016), while the depth loss is the Scale-Invariant Logarithmic (SILog) loss (Eigen et al., 2014). The model is trained on a collection of four datasets: two synthetic datasets, Structured3D (Zheng et al., 2020) and Deep360 (Li et al., 2022), and two real-world datasets, Matterport3D (Chang et al., 2017) and Stanford2D3D (Armeni et al., 2017). Through this training regimen, our feed-forward 3DGS model demonstrates precise geometric modeling capabilities and robust generalization across indoor, outdoor, and even stylized scenes.

### 3.3 3D SCAFFOLD GUIDED NOVEL VIEW SYNTHESIS

In the final stage of our pipeline, we leverage the 3D geometric scaffold to generate a fully explorable 3D scene. In particular, we propose to transform the task of novel view synthesis from a single view to the problem of synthesis conditioned on the set of anchor views:

$$p\left(\mathbf{I}^{\text{tgt}} \mid \mathbf{I}^{\text{anchor}}, \mathbf{p}^{\text{anchor}}, \mathbf{p}^{\text{tgt}}\right). \tag{2}$$

However, the above formulation remains limited since the anchor views are all observations from a single point in the space, and they lack the explicit scale and geometric information required for robust 3D understanding. Our 3D geometric scaffold, with its precise geometric modeling capabilities,

overcomes this limitation by enabling the rendering of novel views from arbitrary viewpoint. These rendered views contain rich geometric and appearance information. Therefore, they can serve as powerful conditions to guide the synthesis of novel views, significantly enhancing their realism and consistency. Although these rendered views may exhibit artifacts or occlusions (e.g., black holes) for large viewpoint changes, they still retain a substantial amount of useful structural information, owing to our model's accurate depth estimation. This insight allows us to further reformulate the synthesis problem as follows:

$$p\left(\mathbf{I}^{\text{tgt}} \mid \mathbf{I}^{\text{anchor}}, \mathbf{p}^{\text{anchor}}, \mathbf{I}^{\text{render}}, \mathbf{p}^{\text{tgt}}\right), \tag{3}$$

where view $\mathbf{I}^{\text{render}}$ is rendered from the scaffold in the camera pose of the target view $\mathbf{I}^{\text{tgt}}$.

**Dual-LoRA Training.** It is a challenging task to manage two distinct types of conditions in the synthesis process: the high-quality anchor views, which offer pristine appearance but are geometrically ambiguous, and the rendered views, which provide strong geometric priors but may contain artifacts. To effectively guide the synthesis using both conditions, we need to process these heterogeneous signals. Inspired by MMDiT (Esser et al., 2024), which uses separate encoders for different modalities, such as text and images, before fusing their features for self-attention, we propose a Dual-LoRA training strategy. Built upon the SEVA architecture (Zhou et al., 2025), our approach employs two different LoRA modules to process the anchor view and the rendered view independently, as shown in Figure 2 (c). The features from both conditions are then integrated with the noisy latent representation through a 3D attention mechanism. Our experiments confirm that this method demonstrates significantly stronger learning capabilities compared to a naive approach of simply concatenating the rendered view with the noise latent.

**Memory Condition.** To ensure temporal and spatial consistency when generating a large number of frames for a continuous 3D scene, we introduce an additional memory condition during inference. This condition is a previously generated frame selected from a memory bank, which has the closest average camera pose to the current target frame. The synthesis problem is thus further refined to:

$$p\left(\mathbf{I}^{\text{tgt}} \mid \mathbf{I}^{\text{anchor}}, \mathbf{p}^{\text{anchor}}, \mathbf{I}^{\text{render}}, \mathbf{I}^{\text{mem}}, \mathbf{p}^{\text{mem}}, \mathbf{p}^{\text{tgt}}\right). \tag{4}$$

This memory-guided approach effectively preserves visual consistency, particularly when synthesizing content in occluded regions.

**Training Data Construction**. To assemble a dataset for supervised training, we perform sparse 3D reconstructions on the DL3DV (Ling et al., 2023) and RealEstate10K (Zhou et al., 2018) datasets using the pre-trained feed-forward 3DGS model MVSplat (Chen et al., 2025b). This strategy is intentionally employed to simulate the artifacts and holes that arise in rendered views when the reconstruction is based on sparse input viewpoints. By using the camera trajectories inherent to these datasets, we sample novel views that exhibit significant viewpoint deviations. Training pairs are subsequently formed, each comprising a ground truth image and its corresponding view rendered from the sparse 3D reconstruction at the identical camera pose.

## 4 EXPERIMENTS

### 4.1 EXPERIMENTAL SETTINGS

**Implementation Details**. In the panorama generation stage, we employ Hunyuan-Pano-DiT (Wang et al., 2025c) as the generator. The feed-forward 3DGS model is trained for 80,000 iterations using the AdamW optimizer. We set the learning rate of the VGGT backbone to 2e-5, and set the learning rate to 2e-4 for all other modules. In the final stage, the 3D scaffold-guided novel view synthesis model is trained for 40,000 iterations using the Adam optimizer based on SEVA (Zhou et al., 2025), with a batch size of 16 and a learning rate of 1.25e-5.

**Experiments Setup**. To more comprehensively evaluate our proposed One2Scene model and demonstrate its effectiveness and advantages, we conduct the following experiments. (1) First, we benchmark our One2Scene model against the SOTA 3D scene generation models in producing high-quality, explorable 3D scenes. (2) Second, we evaluate the key component of our One2Scene model, i.e., the feed-forward 360° reconstruction network, by comparing its quality, efficiency, and geometric accuracy with the SOTA methods. Its depth estimation performance is also evaluated on standard panorama depth estimation benchmarks. (3) Third, we conduct a series of ablation studies to dissect the effectiveness of our design of One2Scene.

**Evaluation Metrics**. We evaluate the quality of our generated scenes across three key dimensions. (1) Visual Fidelity. We measure visual quality using two no-reference image quality assessment metrics: NIQE (Mittal et al., 2012) and Q-Align (Wu et al., 2023). (2) Semantic Consistency. We measure the semantic consistency between the initial image and the novel views using CLIP-I score (Hessel et al., 2021). (3) Geometric Consistency. We evaluate geometric stability by first estimating the camera poses of the generated views with a pre-trained VGGT model. These estimated poses are then benchmarked against the ground-truth camera trajectories to compute Rotation Error (RotError) (He et al., 2024), Camera Motion Consistency (CamMC) (Wang et al., 2024c), and Translation Error (TransError) (He et al., 2024). More details of our evaluation protocol are provided in **??**.

## 4.2 MAIN RESULTS

### 4.2.1 EXPLORABLE 3D SCENE GENERATION

To establish a rigorous evaluation protocol in the absence of a standard benchmark for explorable 3D scene generation, we adapt the WorldScore benchmark (Duan et al., 2025), which is originally proposed for short-sequence 3D scene evaluation. To ensure a comprehensive assessment, we sample 40 scenes spanning four diverse static scene categories: indoor-real, indoor-stylized, outdoor-real, and outdoor-stylized (10 per category). This diverse benchmark allows us to thoroughly test the robustness and quality of the generated 3D scenes from single-view inputs.

**Results**. We compare One2Scene with DreamScene360 (Zhou et al., 2024), WonderJourney (Yu et al., 2023), VMem (Li et al., 2025a) and SEVA (Zhou et al., 2025). Quantitative results are reported in Table 1. For methods that accept camera-conditioned novel view synthesis, we additionally evaluate geometric consistency. Since DreamScene360 and WonderJourney do not produce fully explorable scenes (as shown in Figure 1), we can only perform qualitative comparisons with VMem and SEVAin, as shown in Figure 3. We also condition VMem and SEVA on the anchor views produced in our One2Scene method, and denote the corresponding methods as VMem+ and SEVA+.

**Semantic and Appearance Consistency**. As demonstrated in Figure 3, SEVA and VMem often hallucinate content in unobserved regions, leading to semantic inconsistencies. Our 3D scaffold, however, preserves global semantic coherence. This advantage is validated by our quantitative results in Table 1: our One2Scene achieves superior NIQE (4.43) and Q-Align (4.13) scores, and its CLIP-I score (89.95) markedly surpasses those of SEVA (87.82) and VMem (75.80).

**Scale Ambiguity and Drift**. As noted by Zhou et al. (2025) in SEVA, the single input image makes SEVA suffer from scale ambiguity issues. This manifests the distortion of object size and physically implausible geometric artifacts, such as cameras penetrating through walls (see Figure 3). Even conditioned on our anchor views, SEVA+ and VMem+ remain unable to effectively resolve the scale drift problem. This fundamental limitation stems from the lack of relative translation information in anchor views, which prevents the model from inferring a unified global scale. In contrast, our method explicitly constructs a 3D scaffold that provides robust scale constraints, effectively mitigating the scale ambiguity issue and producing physically plausible results.

**Geometric Stability**. Existing methods often struggle to maintain long-term geometric stability. SEVA, for example, lacks a persistent geometric representation, causing inconsistent reconstructions in loop-closure scenarios (e.g., frame 78 vs. 255 in Figure 3). VMem attempts to enforce consistency via online reconstruction with CUT3R, but this strategy is highly susceptible to a vicious cycle of error accumulation: generated low-quality frames destroy the geometry, which in turn provide wrong guidance for subsequent frames, leading to catastrophic failure. In contrast, our pre-built 3D scaffold provides a stable geometric prior, effectively preventing error propagation. This advantage is substantiated by the quantitative results: our method achieves a score of 0.389 in CamMC, significantly outperforming VMem (0.998, see Table 1).

The above results highlight the superiority of our three-stage design of One2Scene, which systematically addresses the global semantic inconsistency, scale ambiguity, and geometric instability. More results can be found in **??** and our anonymous project page. Camera

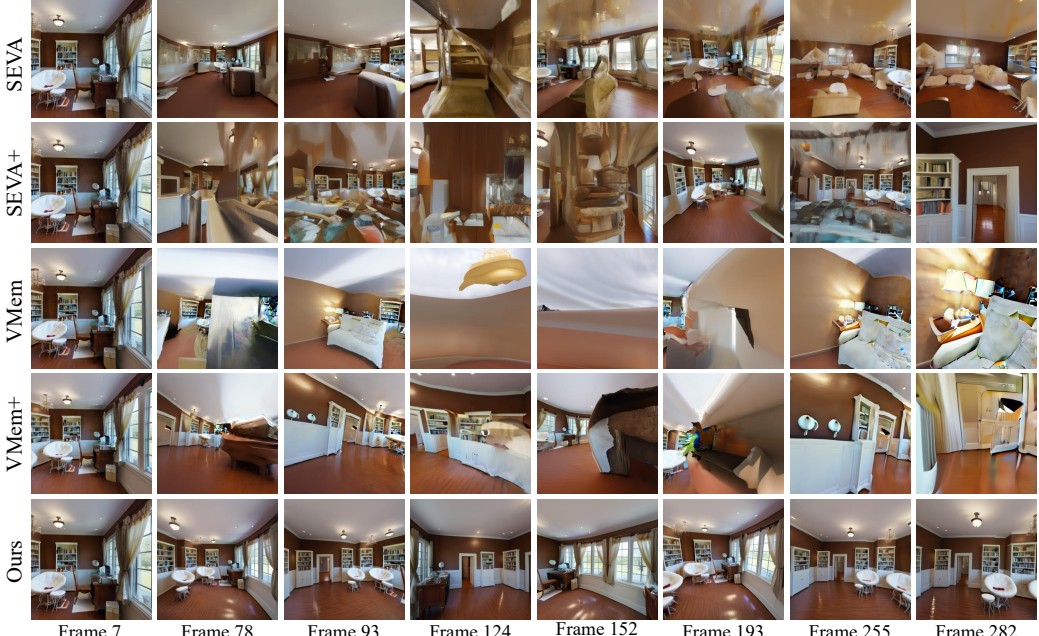

Figure 3: **Qualitative comparison.** Our method retains compelling visual quality and generates plausible continuations of the scene, even under large viewpoint change.

Table 1: Quantitative comparisons for 3D scene generation.

| Methods | NIQE↓ | Q-Align↑ | CLIP-I↑ | TransErr↓ | RotErr↓ | CamMC↓ |
|---|---|---|---|---|---|---|
| DreamScene360 (Zhou et al. (2024)) | 8.40 | 1.91 | 74.24 | - | - | - |
| WonderJourney (Yu et al. (2023)) | 4.97 | 3.02 | 77.92 | - | - | - |
| SEVA (Zhou et al. (2025)) | 4.53 | 3.20 | 87.82 | 0.460 | 0.165 | 0.558 |
| SEVA (Zhou et al. (2025)) + Anchor | 4.45 | 3.45 | 88.70 | 0.422 | 0.116 | 0.460 |
| VMem (Li et al. (2025a)) | 6.86 | 2.95 | 75.80 | 0.573 | 0.569 | 0.998 |
| VMem (Li et al. (2025a)) + Anchor | 5.23 | 3.04 | 81.33 | 0.613 | 0.426 | 0.887 |
| **One2Scene (Ours)** | **4.43** | **4.13** | **89.95** | **0.326** | **0.107** | **0.389** |

### 4.2.2 FEED-FORWARD 360° RECONSTRUCTION

This section validates the core advantages of our feed-forward 3DGS network, a cornerstone of our pipeline. We demonstrate its superiority in reconstruction quality, computational efficiency, and geometric accuracy compared to SOTA methods.

**Reconstruction Quality**. We conduct a direct comparison with the SOTA method, AnySplat (Jiang et al., 2025). Since both methods are extensions of the VGGT model, this shared foundation ensures a fair evaluation. As shown in Figure 4, AnySplat's reconstruction fails with only 6 sparse views. This is because it predicts an erroneous depth map, which results in a distorted geometric scene. Even when 20 densely tangent patches with substantial overlap are projected from a panorama, its performance remains sub-par, suffering from severe artifacts in drastic viewpoint changes. In stark contrast, our model constructs a high-quality and robust 3D geometric scaffold even from sparse inputs. Although large rotations can introduce minor local artifacts due to occlusion, the underlying geometric foundation remains stable, providing crucial priors for the subsequent generation task.

Table 2: Comparison on the 3D scene generation performance by replacing our feed-forward 360° reconstruction network with AnySplat.

| Methods | NIQE↓ | Q-Align↑ | CLIP-I↑ | TransErr↓ | RotErr↓ | CamMC↓ |
|---|---|---|---|---|---|---|
| AnySplat (Jiang et al., 2025) | 4.96 | 3.61 | 81.96 | 0.332 | 0.367 | 0.616 |
| **Ours** | **4.43** | **4.13** | **89.95** | **0.326** | **0.107** | **0.389** |

Table 3: Comparison of depth estimation on Matterport3D and Stanford2D3D datasets.

| Methods | Matterport3D | | | | Stanford2D3D | | | |
|---|---|---|---|---|---|---|---|---|
| | $AbsRel\downarrow$ | $\delta_1\uparrow$ | $\delta_2\uparrow$ | $\delta_3\uparrow$ | $AbsRel\downarrow$ | $\delta_1\uparrow$ | $\delta_2\uparrow$ | $\delta_3\uparrow$ |
| BiFuse (Wang et al., 2020) | 0.2048 | 84.52 | 93.19 | 96.32 | 0.1209 | 86.60 | 95.80 | 98.60 |
| UniFuse (Jiang et al., 2021) | 0.1063 | 88.97 | 96.23 | 98.31 | 0.1114 | 87.11 | 96.64 | 98.82 |
| HoHoNet (Sun et al., 2020)) | 0.1488 | 87.86 | 95.19 | 97.71 | 0.1014 | 90.54 | 96.93 | 98.86 |
| BiFuse++ (Wang et al., 2022) | − | 87.90 | 95.17 | 97.72 | − | 87.83 | 96.49 | 98.84 |
| ACDNet (Zhuang et al., 2022) | 0.1010 | 90.00 | 96.78 | 98.76 | 0.0984 | 88.72 | 97.04 | 98.95 |
| PanoFormer (Shen et al., 2022) | 0.0904 | 88.16 | 96.61 | 98.78 | 0.1131 | 88.08 | 96.23 | 98.55 |
| HRDFuse (Ai et al., 2023) | 0.0967 | 91.62 | 96.69 | 98.44 | 0.0935 | 91.40 | 97.98 | 99.27 |
| EGFormer (Yun et al., 2023) | 0.1473 | 81.58 | 93.90 | 97.35 | 0.1528 | 81.85 | 93.38 | 97.36 |
| Elite360D (Ai & Wang, 2024) | 0.1115 | 88.15 | 96.46 | 98.74 | 0.1182 | 88.72 | 96.84 | 98.92 |
| Depth Anywhere (Wang & Liu, 2024) | 0.0850 | 91.70 | 97.60 | 99.10 | 0.1180 | 91.00 | 97.10 | 98.70 |
| **Ours (Zero-shot)** | 0.1070 | 88.97 | 96.51 | 98.61 | 0.0675 | 95.20 | 98.53 | 99.30 |
| **Ours (Finetune)** | **0.0391** | **98.09** | **99.41** | **99.74** | **0.0444** | **96.95** | **98.85** | **99.44** |

The importance of our scaffold is further confirmed by the experiment in Table 2: replacing our reconstruction module with AnySplat causes a significant degradation in final generation quality.

**Computational Efficiency**. Using six sparse views, our model reconstructs a high-quality scaffold in 0.5 seconds on an H20 GPU, marking a $5.6\times$ speedup over AnySplat, which relies on a dense view set and requires 2.8 seconds. The inference time is further slashed to only 0.1 seconds when using a more powerful NVIDIA H100 GPU.

**Accurate Depth Estimation**. To quantitatively assess the geometric accuracy of our model, we evaluate its depth estimation performance against SOTA methods on the Matterport3D and Stanford2D3D datasets. As detailed in Table 3, the results are compelling: our model, when applied in a zero-shot setting, surpasses all compared approaches on the Stanford2D3D dataset. This result indicates that our method effectively inherits and transfers geometric priors from the foundational VGGT model. Furthermore, when our model is fine-tuned on the Matterport3D and Stanford2D3D datasets, it demonstrates exceptional performance, boosting the AbsRel metric by over 50%. This further underscores the powerful geometric modeling capabilities of our reconstruction model.

## 4.3 ABLATIONS AND ANALYSIS

Given limited space, we provide comprehensive ablation studies in the Appendix, featuring in-depth analyses of our Dual-LoRA training methodology, memory condition mechanism, and bidirectional fusion module (see **??**). We also provide detailed quantitative evaluation results for our generation model on the DL3DV dataset (see **??**).

## 5 CONCLUSION AND LIMITATIONS

In this paper, we introduced One2Scene, a novel and effective framework for generating fully explorable 3D scenes from a single image. We addressed the critical challenge of geometric distortion and artifact generation in existing methods when there were large viewpoint changes. Our core contribution lied in the decomposition of this ill-posed problem into three tractable subtasks: initializing sparse anchor views via a panorama generator, lifting them into an explicit and geometrically reliable 3D scaffold by a feed-forward GS network, and finally, leveraging the scaffold as a strong prior for photorealistic novel view synthesis. Our extensive experiments validated that One2Scene substantially outperformed state-of-the-art methods in explorable 3D scene generation.

**Limitations.** While our approach significantly improves 3D consistency across long sequences and large viewpoint changes, the generated views may contain subtle inconsistencies. Similar to CAT3D (Gao et al., 2024), we can further enhance geometric consistency through post-reconstruction processing. Please see the "Result Gallery" on our anonymous project page. In future work, we plan to construct larger-scale datasets to further improve our model's performance and robustness.

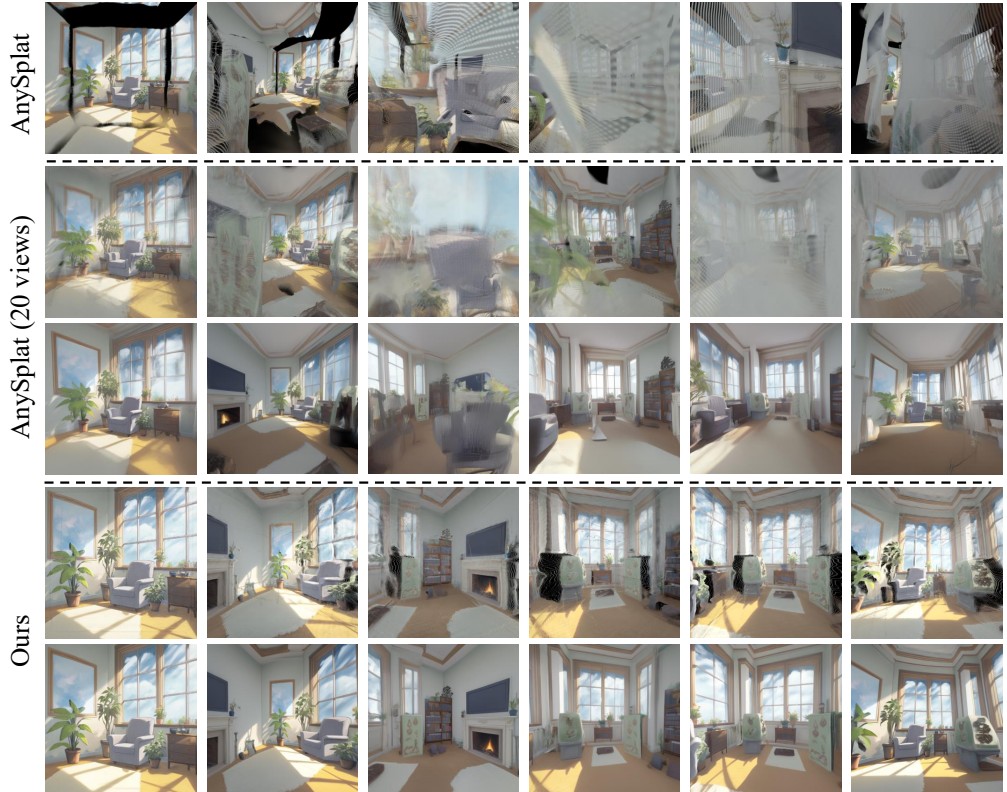

Figure 4: Ablation study on reconstruction performance. We compare the 3D scene generation quality by replacing our feedforward network with AnySplat. Top row: reconstruction results. Bottom row: generation results using our model.

## 6  ETHICS STATEMENT

This research does not involve human participants or the collection of sensitive personal information. All datasets utilized in this study are employed in strict accordance with their respective licensing agreements and terms of use.

The proposed methodology is designed exclusively for academic research and scientific advancement. While we do not anticipate direct harmful applications, we recognize the potential for misuse if deployed without appropriate ethical considerations and safety measures. We advocate for the responsible application of our research contributions, emphasizing the importance of fairness, transparency, and adherence to applicable legal frameworks.

## 7  REPRODUCIBILITY STATEMENT

We have implemented comprehensive measures to facilitate the reproducibility of our research findings. The main manuscript provides thorough documentation of our proposed framework, including detailed descriptions of the model architecture, dataset preprocessing methodologies, and algorithmic implementations. Complete hyperparameter configurations and training protocols are explicitly specified to enable independent replication of our results.

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
