# ONE2SCENE: GEOMETRIC CONSISTENT EXPLORABLE 3D SCENE GENERATION FROM A SINGLE IMAGE

**Pengfei Wang**[*], **Liyi Chen**[*], **Zhiyuan Ma, Yanjun Guo, Guowen Zhang, Lei Zhang**[†]
The Hong Kong Polytechnic University
`pengfei.wang@connect.polyu.hk`, `cslzhang@comp.polyu.edu.hk`
[*]Equal contribution. [†]Corresponding author.
Project page: `https://one2scene5406.github.io/`

## A APPENDIX

We provide the following materials in this appendix:

- Appendix A.1: Detailed evaluation protocol.
- Appendix A.2: Details about cube projection.
- Appendix A.3: Details about bidirectional fusion module.
- Appendix A.4: Ablation study and analysis.
- Appendix A.5: More NVS results on DL3DV.
- Appendix A.6: More qualitative results.
- Appendix A.7: Declaration of LLM assistance.

### A.1 EVALUATION PROTOCOL

To assess the quality of our generated scenes, we evaluate them across three key aspects: visual quality, input-output alignment, and geometric consistency.

For visual quality, we use two no-reference image quality assessment (NR-IQA) metrics. The first is NIQE (Mittal et al., 2012), where a lower score indicates that the image's statistics are more similar to a natural image. The second is Q-Align (Wu et al., 2023), a state-of-the-art model where a higher score signifies better perceptual quality.

For input-output alignment, we use the CLIP-I score (Hessel et al., 2021) to measure the semantic similarity between the generated images and the single input image. A higher score means the content and style are better preserved.

For geometric consistency, we evaluate how accurately the generated camera trajectory matches the ground truth. Our process is as follows: we sample a frame for every 10 frames from the generated sequence, estimate their camera poses using a pre-trained VGGT model (Wang et al., 2025), and then compare these estimated poses to the ground-truth poses used for generation. This comparison is quantified using three metrics: RotError, TransError, and CamMC. To ensure a fair comparison, all methods are tested on the same set of camera trajectories, which are combinations of linear movements (move forward/backward/left/right) and curvilinear movements (orbit, lemniscate). These metrics are defined as follows:

**RotError (He et al., 2024)**. It measures the average per-frame rotation error between the estimated rotation $\tilde{R}_i$ and the ground-truth rotation $R_i$:

$$\text{RotErr} = \frac{1}{n} \sum_{i=1}^{n} \arccos \frac{\text{tr}(\tilde{R}_i R_i^{\text{T}}) - 1}{2}.$$

**TransError (He et al., 2024)**. It measures the average per-frame position error, calculated as the L2 distance between the estimated translation $\tilde{T}_i$ and the ground-truth translation $T_i$:

$$\text{TransErr} = \frac{1}{n} \sum_{i=1}^{n} \left\| \tilde{T}_i - T_i \right\|_2.$$

**CamMC (Wang et al., 2024).** It provides a single score for the average overall pose error by computing the Frobenius norm of the difference between the estimated and ground-truth 3x4 pose matrices:

$$\text{CamMC} = \frac{1}{n} \sum_{i=1}^{n} \left\| \left[ \tilde{R}_i | \tilde{T}_i \right] - [R_i | T_i] \right\|_F.$$

For all geometric error metrics, a lower value indicates better performance.

## A.2 Details about Cube Projection

For equirectangular to cube E2C) projection, the field-of-view (FoV) of each cube face is equal to 90 degrees; each face can be considered as a perspective camera whose focal length is $w/2$, and all faces share the same center point in the world coordinate. Since the six cube faces share the same center point, the extrinsic matrix of each camera can be defined by a rotation matrix $R_i$. $p$ is then the pixel on the cube face:

$$p = K \cdot R_i^T \cdot q, \tag{1}$$

where

$$q = \begin{bmatrix} q_x \\ q_y \\ q_z \end{bmatrix} = \begin{bmatrix} sin(\theta) \cdot \cos(\phi) \\ \sin(\phi) \\ \cos\theta \cdot \cos\phi \end{bmatrix}, K = \begin{bmatrix} w/2 & 0 & w/2 \\ 0 & w/2 & w/2 \\ 0 & 0 & 1 \end{bmatrix}, \tag{2}$$

where $\theta$ and $\phi$ are longitude and latitude in equirectangular projection and $q$ is the position in Euclidean space coordinates.

While the 90° FoV model is mathematically exact for a perfect cube, it can introduce rendering artifacts at the seams between adjacent faces. To resolve this, we expand the field-of-view slightly, for instance to 95°. This modification ensures that each cube face captures a small, overlapped region from its neighbors. The projection methodology remains the same, but the camera's intrinsic matrix must be recalculated.

The relationship between focal length $f$, image width $w$, and FoV is given by $f = (w/2)/\tan(\text{FoV}/2)$. For a 95° FoV, the new focal length, denoted by $f'$, is:

$$f' = \frac{w/2}{\tan(95°/2)} = \frac{w/2}{\tan(47.5°)}. \tag{3}$$

This results in a modified intrinsic matrix, $K'$, where the focal length term $w/2$ is replaced by $f'$:

$$K' = \begin{bmatrix} \frac{w/2}{\tan(47.5°)} & 0 & w/2 \\ 0 & \frac{w/2}{\tan(47.5°)} & w/2 \\ 0 & 0 & 1 \end{bmatrix}. \tag{4}$$

The final projection equation using the improved model is:

$$p = K' \cdot R_i^T \cdot q. \tag{5}$$

This adjustment, while minor, is critical for producing high-quality, artifact-free cubemaps suitable for production rendering environments. The definitions of $q$ and $R_i$ remain unchanged.

The inverse transformation, Cube to Equirectangular (C2E) projection, which is used to project features from the cube faces back to the panoramic view, is achieved by mathematically reversing this projection process. This robust projection method is essential for the bidirectional feature exchange in our model.

## A.3 Details about Bidirectional Fusion Module

The performance of traditional multi-view models, such as VGGT that relies on dense overlap, degrades significantly when faced with extremely sparse correspondences resulting from a mere 2.5-degree overlap between anchor views. To address this issue, we introduce an innovative modification to the VGGT architecture, which aims to explicitly enhance cross-view consistency, thereby improving the robustness of depth estimation. Specifically, we integrate a Bidirectional Fusion Module into the

pre-trained DPT head to promote cross-view depth consistency. The core principle of this module is to establish geometric correspondences across views while preserving the unique, high-fidelity details inherent to each individual view.

The module commences with the feature maps $\{\mathbf{F}_i\}_{i=1}^6$ extracted from the six anchor views. To effectively process the overlapping regions, we first introduce a C2E transformation module. As detailed in Appendix A.2, the C2E transformation leverages strict geometric projection principles to seamlessly project and aggregate the features from the six discrete cube views into a unified equirectangular latent space via differentiable bilinear sampling.

Subsequently, a lightweight convolutional layer, $\mathbf{H}_c$, is applied to this aggregated global feature map. Its purpose is to smooth the boundaries between the projected views and fuse their information, forming a globally consistent feature representation, $\mathbf{F}_e$. This step can be conceptualized as a process that information from all views is aggregated to build a consensus representation. This forward fusion process is formulated as:

$$\mathbf{F}_e = \mathbf{H}_c(\text{C2E}(\{\mathbf{F}_i\}_{i=1}^6)). \tag{6}$$

Next, to propagate this global consistency information back to each individual view, we perform an inverse process. Through an E2C transformation, the fused global feature $\mathbf{F}_e$ is re-projected into the coordinate spaces of the six original anchor views.

Finally and crucially, rather than directly replacing the original features with this global information, we employ a residual connection to add it to the original feature map $\mathbf{F}_i$, yielding the updated view-specific feature $\mathbf{F}_i'$:

$$\mathbf{F}_i' = \mathbf{F}_i + \text{E2C}(\mathbf{F}_e). \tag{7}$$

The elegance of this "local-to-global-to-local" bidirectional mechanism lies in its dual function: the C2E/E2C transformations are responsible for aligning features in overlapping regions to enforce geometric consistency, while the residual connection ensures that the model retains and utilizes the original, high-fidelity details from each view. In this manner, our module effectively strengthens cross-view constraints while preventing the loss of view-specific information that can occur with forced fusion.

### A.4 ABLATION STUDY AND ANALYSIS

**Effectiveness of Dual-LoRA Training.** We first compare our Dual-LoRA training against the common channel-wise concatenation method. As shown in Figure A1, our model exhibits superior generation quality, no matter with and without the memory condition. This is because our Dual-LoRA approach can better leverage the two conditions of varying quality. The results in Table A1 further confirm that Dual-LoRA achieves better visual quality and geometric consistency.

**Effectiveness of Memory Condition.** We then analyze the impact of incorporating an additional memory condition at inference time. Although the quantitative results in Table A1 do not show a significant improvement, we observe a clear qualitative benefit. As highlighted by the colored boxes in Figure A1, this condition helps our model maintain better multi-view consistency, especially in occluded regions requiring significant content synthesis.

**Effectiveness of Bidirectional Fusion Module.** Our baseline approach directly applies VGGT for multi-view consistent depth estimation. However, due to the extremely sparse overlap between anchor views in panoramic scenarios, VGGT struggles to handle such conditions, resulting in significant performance degradation compared to geometric estimation tasks with larger overlaps. We fine-tune VGGT on panoramic images without any architectural modifications, which leads to noticeable performance improvements but still exhibits seaming artifacts at view boundaries.

Our proposed Bidirectional Fusion (BF) module substantially alleviates the geometric inconsistencies at edges. The BF module leverages complementary Cubemap-to-Equirectangular (C2E) and Equirectangular-to-Cubemap (E2C) transformations to establish robust geometric correspondences through residual connections. This bidirectional information flow enables the model to better handle the sparse overlap challenge inherent in panoramic depth estimation. As demonstrated in Table A2, the integration of the BF module yields significant performance improvements across both datasets, with notable gains in accuracy metrics such as reduced AbsRel error and increased $\delta_1$, $\delta_2$ and $\delta_3$,

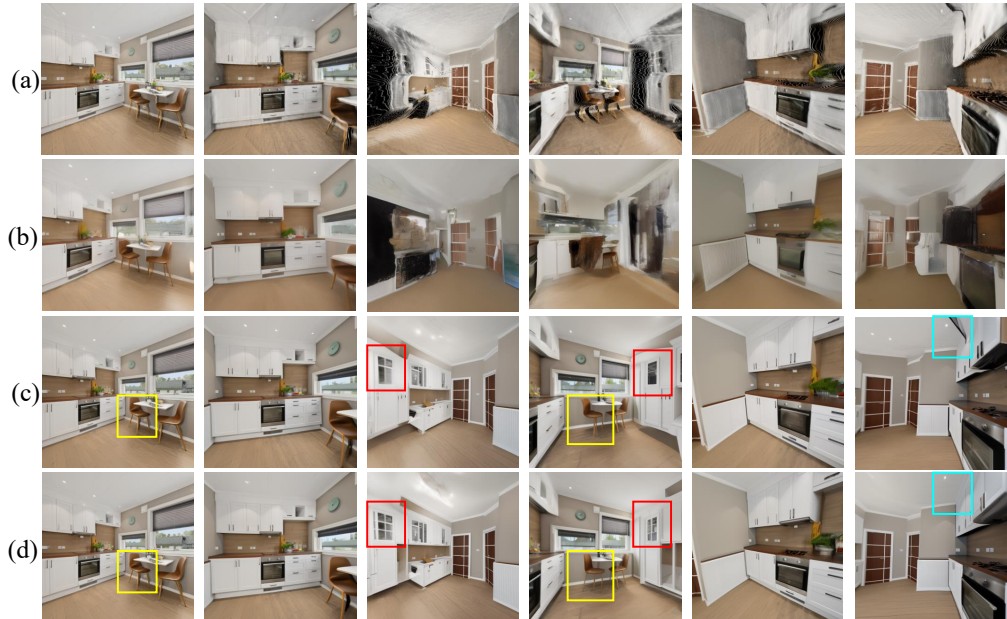

Figure A1: Qualitative comparison for the ablation study. (a) Render views from our 3D scaffold. (b) Naive concatenation baseline. (c) Ours (Dual-LoRA training only). (d) Ours (Full model with memory condition).

Table A1: Ablation study on 3D scaffold guided novel view synthesis.

| Methods | NIQE↓ | Q-Align↑ | CLIP-I↑ | TransErr↓ | RotErr↓ | CamMC↓ |
|---|---|---|---|---|---|---|
| Naive Concat. | 5.04 | 3.41 | 85.30 | 0.481 | 0.260 | 0.655 |
| Dual-LoRA Training | 4.42 | 4.10 | 89.51 | 0.326 | 0.119 | 0.401 |
| + Memory Condition | 4.43 | 4.13 | 89.95 | 0.326 | 0.107 | 0.389 |

Table A2: Effectiveness of BF module. Zero-shot quantitative comparison on Matterport3D and Stanford2D3D datasets.

| Methods | Matterport3D | | | | Stanford2D3D | | | |
|---|---|---|---|---|---|---|---|---|
| | $AbsRel$↓ | $\delta_1$↑ | $\delta_2$↑ | $\delta_3$↑ | $AbsRel$↓ | $\delta_1$↑ | $\delta_2$↑ | $\delta_3$↑ |
| Baseline | 0.1576 | 78.82 | 93.20 | 96.15 | 0.1497 | 81.99 | 93.53 | 97.88 |
| w/o BF | 0.1204 | 86.28 | 95.36 | 97.45 | 0.0797 | 94.31 | 97.42 | 98.85 |
| w BF | 0.1070 | 88.97 | 96.51 | 98.61 | 0.0675 | 95.20 | 98.53 | 99.30 |

confirming the effectiveness of our approach in addressing multi-view consistency challenges in panoramic depth estimation.

## A.5 NVS RESULTS ON DL3DV

**Competing Method**. Our primary competing method is MVSplat360 (Chen et al., 2024), a state-of-the-art method capable of refining rendered views. To ensure a direct and fair comparison, we strictly adhere to the evaluation protocol established for the DL3DV (Ling et al., 2023) dataset, as utilized by the competing method.

**Quantitative Results**. As detailed in Table A3, our method demonstrates superior performance over MVSplat360 across all evaluation metrics. Specifically, our method achieves a PSNR of 17.35 (+0.98) and an FID of 116.84 (-1.48). Furthermore, we observe substantial reductions in both LPIPS (0.343) and DIST (0.181) indices, indicating superior perceptual similarity and geometric accuracy,

Table A3: The NVS numerical comparison on the DL3DV (Ling et al., 2023) dataset.

| Methods | PSNR (↑) | SSIM (↑) | LPIPS (↓) | DIST (↓) | FID (↓) |
|---|---|---|---|---|---|
| PixelSplat | 15.32 | 0.422 | 0.517 | 0.374 | 139.75 |
| MVSplat | 15.94 | 0.441 | 0.459 | 0.282 | 73.91 |
| MVSplat360 | 16.37 | 0.453 | 0.439 | 0.238 | 18.32 |
| Ours | **17.35** | **0.506** | **0.343** | **0.181** | **16.84** |

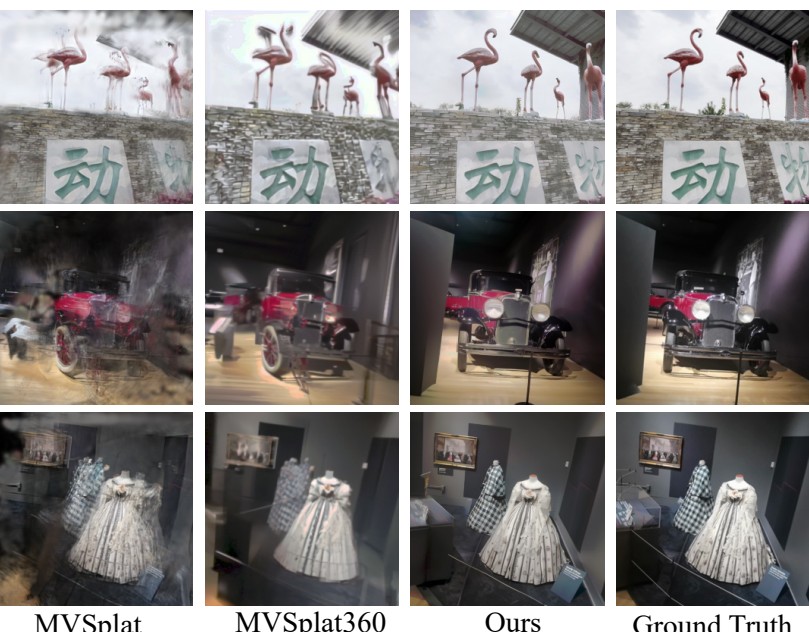

MVSplat          MVSplat360          Ours          Ground Truth

Figure A2: Visual comparison with existing SOTA methods on DL3DV.

respectively. Collectively, these quantitative improvements underscore our method's enhanced effectiveness in leveraging auxiliary views to synthesize more accurate and high-fidelity novel views.

**Qualitative Results**. The qualitative comparisons presented in Figure A2 visually corroborate our quantitative findings. Our method consistently generates sharper and more structurally coherent scenes, showcasing an effective use of information from auxiliary views. In contrast, the results from MVSplat360 frequently exhibit noticeable artifacts and structural distortions, particularly when synthesizing views with large camera pose changes.

## A.6 MORE QUALITATIVE RESULTS

In this section, we provide more qualitative results to further support the claims presented in the main paper. We showcase a broader range of visual comparisons against baseline methods across diverse and challenging scenes, including indoor, outdoor, and stylized scenes. These examples serve to visually corroborate the quantitative improvements reported in the main paper, highlighting our method's superior performance in generating explorable 3D scenes.

We present side-by-side visualizations to compare our method, One2Scene, against key competitors: VMem and SEVA. Consistent with the main paper, we also include results for their '+' variants (VMem+ and SEVA+), which are conditioned on our generated anchor views. These comparisons, as shown from Figure A3 to Figure A7, further demonstrate the superior performance of our method in terms of visual fidelity, 3D geometric consistency, and the effective mitigation of scale ambiguity artifacts in previous methods.

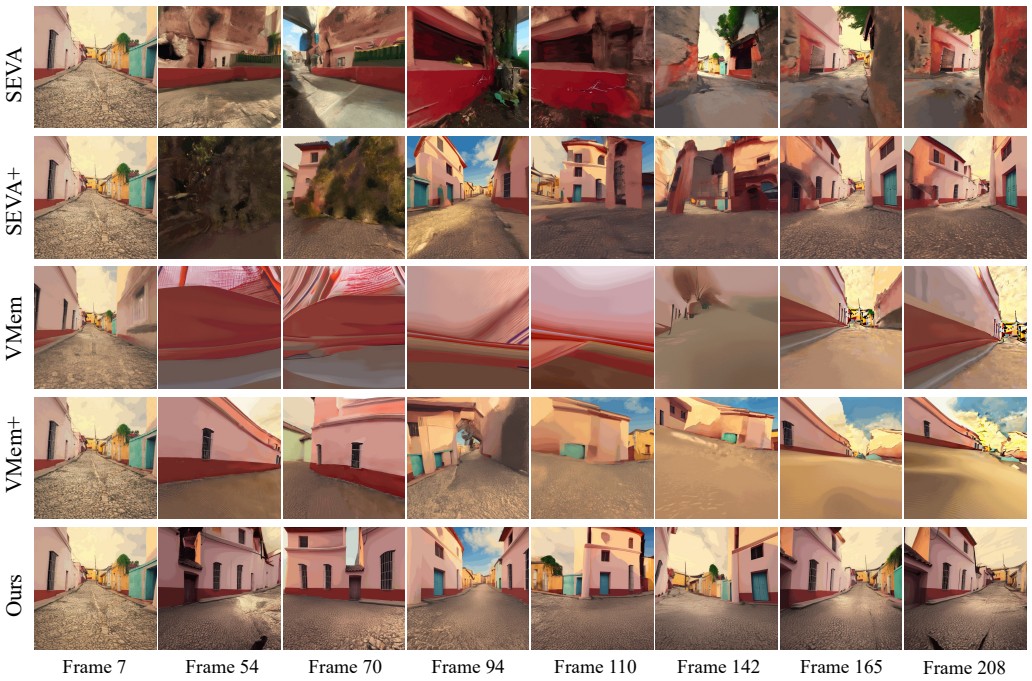

Figure A3: Qualitative comparison between One2Scene and SOTA methods.

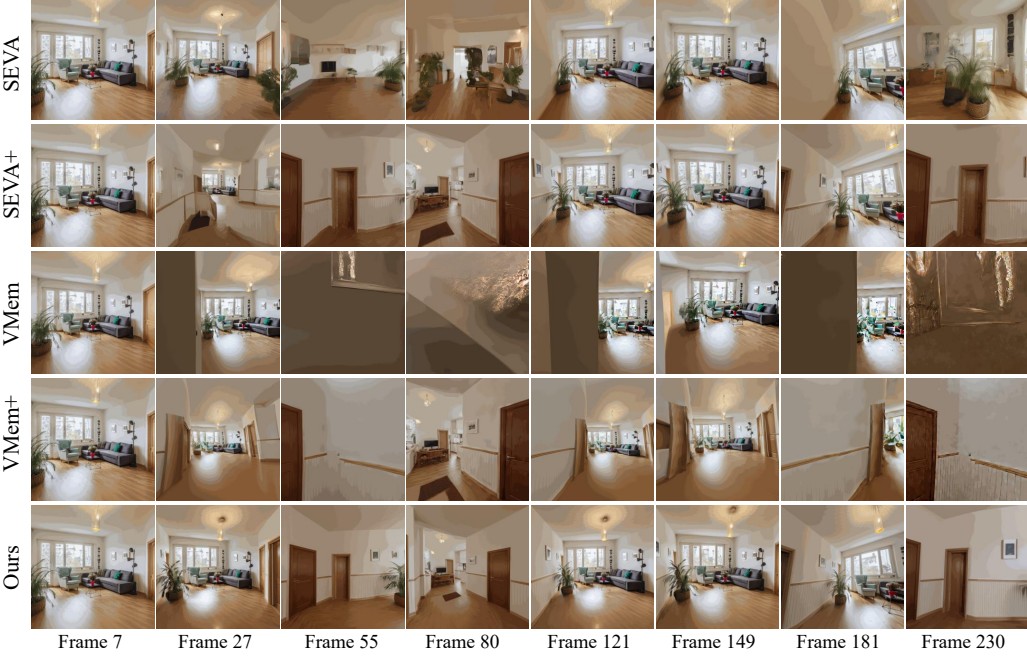

Figure A4: Qualitative comparison between One2Scene and SOTA methods.

## A.7 DECLARATION OF GENERATIVE AI ASSISTANCE

During the preparation of this manuscript, we utilized Gemini-2.5-Pro to assist in improving its linguistic quality. Specifically, after completing the initial draft, we provided the model with selected passages to obtain suggestions for grammar, clarity, and conciseness. All AI-assisted revisions were

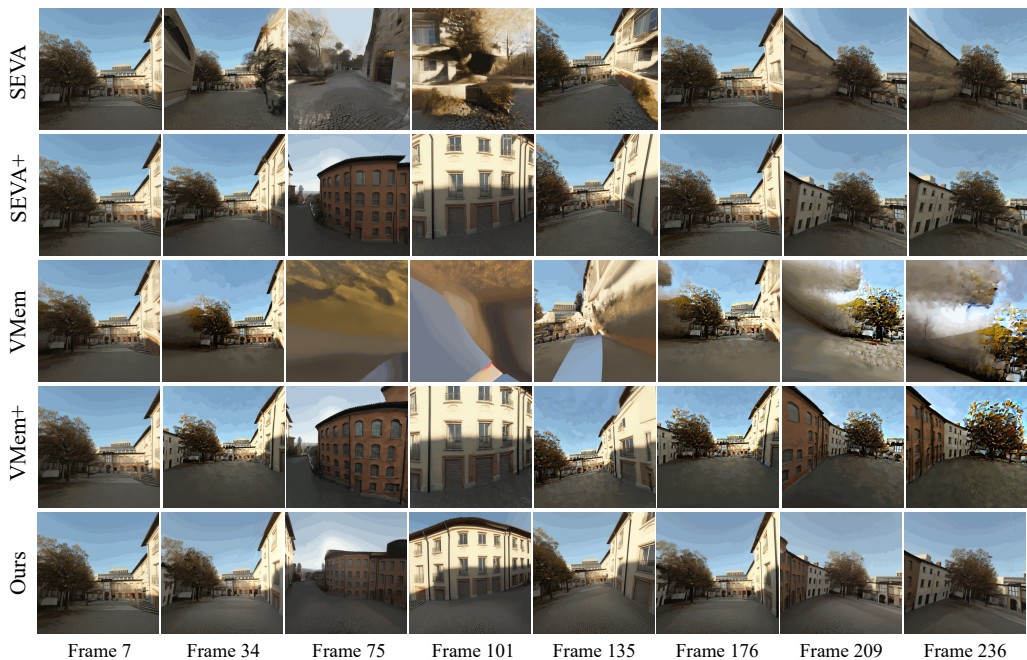

Figure A5: Qualitative comparison between One2Scene and SOTA methods.

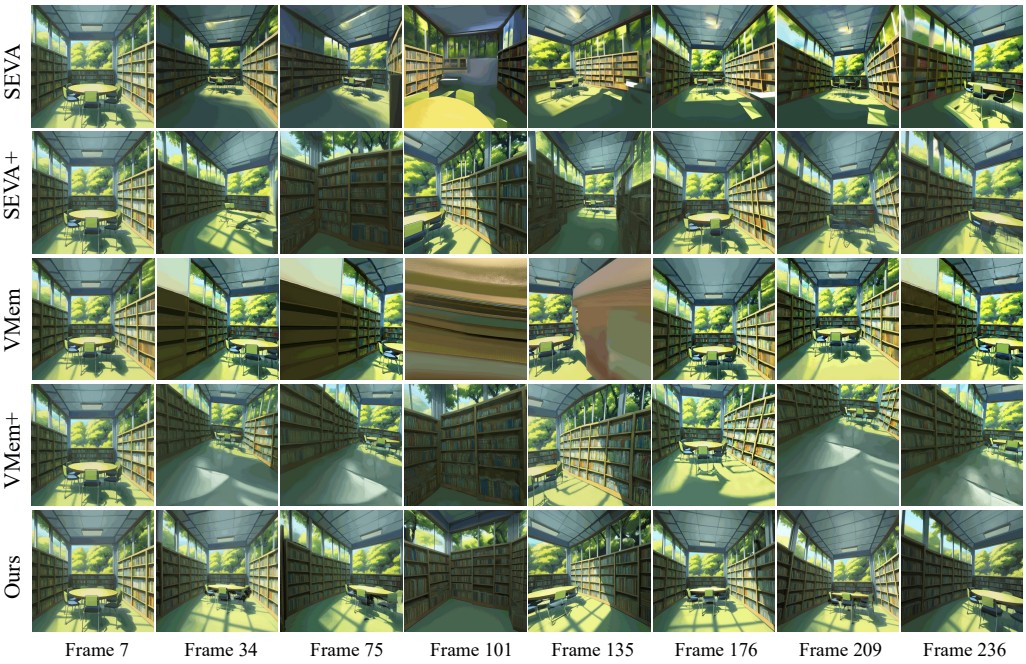

Figure A6: Qualitative comparison between One2Scene and SOTA methods.

rigorously reviewed and edited by the authors, who assume full responsibility for the final accuracy and scholarly appropriateness of the content.

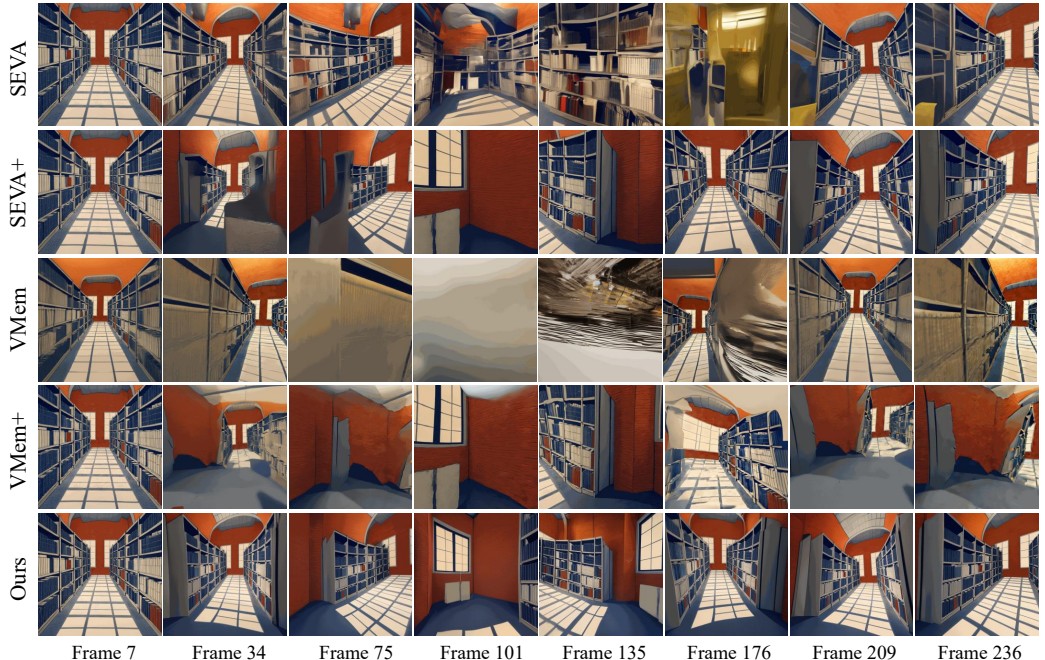

Figure A7: Qualitative comparison between One2Scene and SOTA methods.