# OpenReview forum: "One2Scene: Geometric Consistent Explorable 3D Scene Generation from a Single Image"
_ICLR.cc/2026/Conference — ICLR 2026 Poster_

### Official Review · Reviewer_7LcG · 2025-10-26

**Soundness:** 3
**Presentation:** 3
**Contribution:** 3
**Rating:** 6
**Confidence:** 4

**Summary:**

The manuscript proposes a framework for generating an explorable 3DGS scene from a single perspective image. For doing that, authors decompose the problems in three independent tasks: 1. they generate a panorama 2. they select 2D anchor views from this panorama and create a 3DGS based scaffold by solving a multi-view stereo matching problem; 3. they use the scaffold for creating novel views at arbitrary camera poses.  They assess the framework in multiple ways: panoramic depth estimation, scaffold reconstruction, and explorable generation. The project website contain additional information and qualitative results.

**Strengths:**

1. The paper is well written and technically solid. Narrative style is convincing, and the manuscript is easy to follow.
2. The problem is really challenging and worth of exploration: 3D content generation from single images and automatic world generation has a strong impact on media creation.
3. The proposed solution appears convincing and well designed; technical details are provided, and authors indicate they will release the code for replicability.
4. The website resource contain a good variety of additional qualitative results: they look impressive and they show that the proposed framework has a strong potential.
5. Assessment: the framework is benchmarked against the most modern SOTA, and the relevant literature is generally discussed and compared. Quantitative and qualitative results are generally convincing.

**Weaknesses:**

1. Practical usage: even if the qualitative results appear impressive, I would expect attempts to go at different level of detail. I would like to see some examples of animations with different level of zoom and showing cluttered areas. For example, I would expect attempts to generate detailed leaves in plants, etc..  In most of cases, the generated scenes are with limited clutter. Also, I would expect to see how it works on the wild, with some casual image. Also, it is importante to clearly state what are the image resolution limitations.
2. Comparison with SOTA: in related work section, I would expect some sentences showcasing in detail how the proposed framework improve current technologies.
3. Link to panoramic scene reasoning: since the proposed framework pass through a panorama generation, and then it does all the subsequent tasks on a panoramic image with depth estimation; I would expect a comparison with methods generating 3DGS or explorable scenes from single panoramic images, like for example Pano2Room
Pu, G., Zhao, Y., & Lian, Z. (2024, December). Pano2room: Novel view synthesis from a single indoor panorama. In SIGGRAPH Asia 2024 Conference Papers (pp. 1-11).
https://github.com/TrickyGo/Pano2Room.  Actually, the strategy presented is quite similar to Pano2Room, and I would expect at minimum a comparative discussion, and probably also a benchmark against this baseline.
4. Minor: for fairness, in qualitative results authors should use the same camera poses for baselines.
line 57:  topicm
lines 199-201: please rephrase it. Lots of methods try to perform depth estimation from single panorama for novel view synthesis, like
Pintore, G., Bettio, F., Agus, M., & Gobbetti, E. (2023). Deep scene synthesis of Atlanta-world interiors from a single omnidirectional image. IEEE Transactions on Visualization and Computer Graphics, 29(11), 4708-4718.
Figure A7: one of them is SEVA+, which one?

**Questions:**

1. How the method works on the wild, with casual images? What are the limitations for usage in a game or VR scenarios? Image resolution, details in extreme zooming, etc. I would expect an extensive assessment in supplementary, and some information on the main manuscript
2. How is the method related to competing baseline, like SAVA+?  How is the related to Pano2Room? I would expect a detailed discussion on how the pipeline differs from this paper, as well as a benchmarked comparison against it.
3. Can you provide additional details about the bidirectional fuse model, since it it a key component in the proposed architecture? Some details about Cube Projection can be shortened, instead.

**Details Of Ethics Concerns:**

No ethics concerns for this manuscript

---

> ### Author Response · Authors · 2025-11-20
>
> We sincerely thank this Reviewer for the thorough and thoughtful review. We are encouraged by the positive assessment of our work, such as "technically solid" and "well designed." We especially appreciate the suggestions to benchmark against Pano2Room and to more thoroughly investigate the practical applicability and "in-the-wild" performance of our method. Based on this reviewer's feedback, we have provided substantial updates to both the manuscript and the project page (https://one2scene5406.github.io).
>
> ---
> **Q1. Practical usage**
>
> Thank you for the insightful suggestion. Generating detailed and cluttered scenes is indeed a common challenge in the task of explorable 3D scene generation. In response to your suggestion, we have included examples using images of varying resolutions and scene complexities sourced from the internet. These results are now available on our project page (https://one2scene5406.github.io).
>
> **Complex Scenes:** When generating complex scenes, while our method maintains overall geometric consistency, the quality of the generated details decreases. This is primarily due to the lack of such data in the training dataset. We plan to extend the dataset in future work to further improve the robustness of our approach. Nonetheless, please note that other compared methods will fail under such complex scenes.
>
> **Detail Generation:** We have showcased two scenarios where the model generates fine details, such as the leaves of plants, through zoom-in operations. These examples demonstrate the model's ability to capture finer details in certain conditions, though there is still room for improvement.
>
> **Limitations and VR/Game Applications:** We clarify that the output resolution is 512x512, following prior work like SEVA for fair comparison. For VR/game applications, our feed-forward scaffold generation (0.1s on H100) is highly promising for real-time use. However, the diffusion-based view synthesis stage is slower and would require further optimization (e.g., via distillation) for better interactive experiences.
>
> ---
> **Q2. Comparison with Pano2Room**
>
> Thank you for the suggestion. We acknowledge that Pano2Room shares a similar starting point with us (generating scenes from a panorama); however, our method is fundamentally different from Pano2Room and offers several key advantages. We have added a detailed discussion in the Related Work section and a new quantitative comparison in Table 3.
>
> Our key advantages are:
>
> **1. Generalization.** Pano2Room is explicitly designed with strong indoor priors and is limited to indoor scenes. In contrast, our One2Scene is a general-purpose model capable of generating both indoor and outdoor scenes in various styles, as shown in our gallery.
>
> **2. Efficiency and Scalability.** Pano2Room relies on a per-scene optimization approach with an inpainting model, where the paper recommends fine-tuning the inpainting model for approximately 30 minutes per new panorama to achieve better performance. In contrast, One2Scene is a feed-forward model that generates the 3D scaffold in only 0.1 seconds on an H100 GPU and demonstrates robust synthesis performance without requiring scene-specific fine-tuning. This makes our approach significantly more practical and scalable for real-world applications.
>
> **3. Geometric and Semantic Consistency.** Pano2Room relies on a navigation-and-inpainting strategy, which is prone to semantic drift and style inconsistency over long trajectories. Our scaffold-guided approach first establishes a globally consistent 3D representation, which acts as a strong prior to ensure geometric accuracy and semantic coherence during novel view synthesis, effectively mitigating these risks.
>
> **4. Superior Geometry Foundation.** Pano2Room leverages an off-the-shelf depth estimator and aligns depths from different views. Instead, we designed and trained a dedicated panoramic depth estimation network that demonstrates stronger generalization and accuracy, providing a more robust geometric foundation for the entire pipeline.
>
> **5. Superior Performance.** To provide a direct comparison, we benchmarked both methods on the indoor-real subset of our evaluation set. As shown in our new Table (below), One2Scene significantly outperforms Pano2Room across all metrics, validating that our scaffold-guided framework produces geometrically and photometrically superior results.
>
> | Methods | NIQE↓ | Q-Align↑ | CLIP-I↑ | TransErr↓ | RotErr↓ | CamMC↓ |
> |---------|-------|----------|---------|-----------|---------|--------|
> | Pano2Room | 4.87 | 3.74 | 87.10 | 0.279 | 0.081 | 0.309 |
> | **One2Scene (Ours)** | **4.64** | **4.06** | **88.52** | **0.253** | **0.066** | **0.288** |

---

> > ### Comment · Reviewer_7LcG · 2025-11-27
> > **Satisfying answers for Q1 and Q2**
> >
> > Thanks for providing extensive answers to my questions, that satisfy my concerns. For Q1, I would expect in the final version at least a  short discussion of limitations and a rationale about practical usage, as explained in the answer. Practitioners need to know what is the current SOTA when it comes to usage in VR scenarios.

---

> ### Author Response · Authors · 2025-11-20
>
> ---
> **Q3. Revised related work**
>
> Thank you for this valuable suggestion. We have substantially revised the related work section to provide a more detailed comparison with current state-of-the-art technologies. Specifically, we now explicitly contrast our method with two key lines of work:
>
> (1) Iterative navigation and inpainting methods (e.g., Pano2Room): We highlight our advantages in global
> scene consistency and generality (indoor/outdoor vs. indoor-only with strong priors).
>
> (2) Pose-conditioned view synthesis methods (e.g., SEVA, CAT3D): We emphasize how our global 3D scaffold prevents the error accumulation and semantic drift common in these sequential approaches, ensuring long-range geometric consistency.
>
> These revisions better articulate the unique contributions of our scaffold-guided framework and its improvements over existing technologies.
>
> ---
> **Q4.  Use of the same camera poses for qualitative comparisons**
>
> Thanks for the insightful question. We confirm that we used exactly the same camera trajectory for all methods in our qualitative comparisons. The visual discrepancy this reviewer noted is, in fact, a direct consequence of the baselines' failure to maintain a consistent scale and geometric structure. This leads to artifacts like camera drifting or passing through walls, even given the same camera poses.  The problem of scale ambiguity is discussed in Section 4.2 of the original paper of SEVA.
> This phenomenon demonstrates a key strength of our method: the explicit 3D scaffold effectively resolves scale ambiguity and enforces geometric constraints, ensuring that the generated views faithfully follow the intended trajectory.
>
> ---
> **Q5.  Disscusion with SEVA+**
>
> Thank you for the question. SEVA+ is a variant of SEVA we proposed to serve as a strong and fair baseline. It is an improved version of SEVA conditioned on the same anchor views as our One2Scene. However, SEVA+ lacks the explicit 3D geometric scaffold proposed in One2Scene. As can be seen from our experimental results, One2Scene significantly outperforms SEVA+ in geometric consistency. This direct comparison effectively validates the effectiveness of our 3D scaffold in generating stable, long-range novel views.
>
>
> ---
> **Q6.  More details on the bidirectional fusion module**
>
> Thank you for this valuable feedback. In response, we have introduced Appendix A.3 to provide a more detailed description of the bidirectional fusion module. We trust that this new section will offer better clarity on our technical implementation.
>
> ---
>
> **Q7. Typos and writing suggestions**
>
> Thank you for pointing out these typos and writing suggestions. We have carefully reviewed and addressed them in the updated manuscript.
>
> • Typo on Line 57: We thank the reviewer for identifying this typo. The typo "topicm" has been corrected to "topic" in the revised manuscript.
>
> • Ambiguity in Figure A7: Thank you for pointing out this ambiguity. The label "SEVA" in the second row has been corrected to "SEVA+" in the revised manuscript.
>
> • Rephrase lines 199-201: We have revised the sentence to:
>
> "Although significant progress has been made in depth estimation from a single panoramic image [1, 2, 3], this task remains highly challenging. A key difficulty lies in the lack of large-scale datasets comparable to those available for perspective images, limiting the generalization ability of panoramic depth estimators."
>
> [1] Pintore, G., Bettio, F., Agus, M., and Gobbetti, E. (2023). Deep scene synthesis of Atlanta-world interiors from a single omnidirectional image. IEEE Transactions on Visualization and Computer Graphics, 29(11), 4708-4718.
>
> [2] Hao Ai, Zidong Cao, Yan-Pei Cao, Ying Shan, and Lin Wang. Hrdfuse: Monocular 360° depth estimation by collaboratively learning holistic-with-regional depth distributions. In Proceedings of the IEEE/CVF Conference on Computer Vision and Pattern Recognition, pp. 13273–13282, 2023.
>
> [3] Ning-Hsu Albert Wang and Yu-Lun Liu. Depth anywhere: Enhancing 360 monocular depth estimation via perspective distillation and unlabeled data augmentation. Advances in Neural Information Processing Systems, 37:127739–127764, 2024.

---

> > ### Comment · Reviewer_7LcG · 2025-11-27
> > **Satisfying answers**
> >
> > Thanks for the efforts. My concerns are solved in a satisfying way.

---

> ### Author Response · Authors · 2025-11-27
>
> Dear Reviewer 7LcG,
>
> Thank you for your precious time in reviewing our paper and your constructive comments. We appreciate if you could let us know whether our responses have addressed your concerns.
>
> Best regards,
>
> Authors of paper \#5406

---

> ### Author Response · Authors · 2025-11-27
>
> Dear Reviewer 7LcG,
>
> Thank you very much for your response and for raising our score to 8! We’re delighted to hear that our reply addressed your concerns and truly appreciate your valuable feedback.
>
> Your suggestion to include a discussion on limitations and practical usage for VR practitioners is excellent. We agree this will greatly enhance the paper. In the final version, we will add a new subsection titled "Limitations and Practical Implications" (or similar). In this section, we will thoroughly discuss the boundaries of our approach and provide a clear rationale regarding its real-world application in VR scenarios, as you recommended. Thank you again for helping us improve our work.
>
> Best regards,
>
> Authors of paper #5406

---

### Official Review · Reviewer_U23k · 2025-10-29

**Soundness:** 2
**Presentation:** 2
**Contribution:** 2
**Rating:** 4
**Confidence:** 3

**Summary:**

This paper explores how to leverage a 3D representation to condition the generation of new viewpoints given a single viewpoint. They propose a pipeline that takes an image, uses an off the shelf model to predict panoramas, then selects anchor views and from this constructs a 3D representation using a trained neuraal network. This is then rendered to the target view and, with the anchoring view, fed as a prior to the diffusion model.

The paper's point is that explicitly having an accurate 3D representation is key to obtaining good, robust, generated views as extreme viewpoints and they aim to demonstrate that their approach of doing so is SOTA.

**Strengths:**

The paper appears to have done comprehensive ablations:
1. Comparing sub parts of their approach (the 3D generation) and the image generation part (by using others' 3D representation), finding improvements in both cases.

2. The use of a fixed 3D representation makes a lot of sense to fix geometric mistakes / ambiguity. By creating an approximation of the whole 3D scene a priori, the authors get around issues taht exist in other works due to the accumulation of errors.

**Weaknesses:**

1. Why no comparison with Cat3D ? This seems like an obvious baseline which also does 1 view to multiple views ? The paper focusses on other methods that, to my understanding, seem to be mroe aimed to handle a trajectory of views (e.g. SEVA / AnySplat) and so may in general be expected to handle more complex scenes. While the author's approach is better than these in this case, they should also compare it to methods with more similar aims (e.g. Cat3D).

2. How does the model fair if you don't do a loop closure setting -- e.g. where you keep moving 'forward' by zooming in on a location? I fear in this case the panorama wouldn't be so useful for maintaining consistency and so is this method really only better than others in this one case for the camera angles ?

3. The authors evaluate their full pipeline on only one dataset -- but surely they could explore others as well ? Such as those in Table 1 in the Cat3D paper ?

**Questions:**

1. For Figure 1, it's quite confusing what I'm looking at -- what was input to the model, what was it tasked to do and what was the output? This is useful to put the results in context.

---

> ### Author Response · Authors · 2025-11-20
>
> We sincerely thank this Reviewer for the thoughtful and critical questions regarding our experimental design, choice of baselines, and the method's robustness, which prompt us to more clearly articulate the scope of our work and provide stronger justifications for our evaluation protocol. Our itemized responses can be found below.
>
> ---
>
> **Q1: Why no comparison with CAT3D?**
>
> We thank this reviewer for this suggestion. CAT3D is an important prior work in novel view synthesis, and we have cited and discussed it in our main paper. However, we did not include CAT3D as a quantitative baseline for the following reasons:
>
> **1. Lack of Public Code and Models:** The code and models for CAT3D are not publicly available, which prevents a fair, protocol-aligned, and reproducible comparison.
>
> **2. SEVA as a More Advanced Successor:** We chose SEVA as our primary baseline because it is a more advanced successor to CAT3D. Please note that SEVA addresses the same problem as CAT3D, and their methodologies are closely related. To generate a trajectory, CAT3D first produces discrete viewpoints and then creates smooth sequences between these adjacent views. SEVA follows this principle, uses CAT3D as a key baseline and demonstrates superior performance than CAT3D. Therefore, including SEVA as a baseline naturally subsumes CAT3D, and benchmarking against SEVA already represents a stricter and more up-to-date comparison.
>
> We genuinely appreciate the reviewer highlighting CAT3D, and we thank you again for pointing out this potentially meaningful baseline. We hope the above explanation clarifies our decision and assures the reviewer that we carefully considered the comparison choices.
>
> ---
>
> **Q2: How does the model fare in non-loop-closure settings?**
>
> Thank you for this insightful question. We would like to emphasize that the core problem our method aims to address is how to ensure geometric consistency under long trajectories and large viewpoint changes—an issue that is widely recognized as highly challenging in the field of 3D scene generation. Existing sequential generation methods, such as autoregressive models, almost inevitably suffer from geometric drift and scene collapse due to the accumulation of errors over long trajectories. Our work is specifically designed to overcome these challenges.
>
> In response to your question regarding non-loop-closure settings, such as continuous "forward motion" or "zoom-in" trajectories, our method remains robust. This robustness does not stem from the use of a panorama itself but instead from the globally consistent 3D scaffold constructed from the panorama. Once this scaffold is established, it serves as a fixed, unified reference for both geometry and appearance. During sequence generation, each frame is anchored to this global scaffold for localization and rendering, effectively mitigating the risk of cumulative drift at the architectural level.
>
> On our project page (https://one2scene5406.github.io), we have included comparisons for continuous "forward motion" trajectories. These results demonstrate that our method achieves significantly better geometric consistency compared to baseline approaches, highlighting the effectiveness of our global scaffold.

---

> ### Author Response · Authors · 2025-11-20
>
> ---
>
> **Q3: Why is the evaluation protocol from CAT3D's Table 1 not applicable?**
>
> Thank you for this suggestion. We would like to respectfully clarify a fundamental difference between our task and CAT3D (Table 1 in the CAT3D paper), as shown in the table below.
>
> The benchmark in CAT3D is for multi-view reconstruction (interpolating novel views from 3, 6, or 9 input images with ground truth available). Our work tackles single-view explorable 3D scene generation (extrapolating a full 360° scene from one image, an ill-posed task without ground truth). Due to this fundamental mismatch in task definitions (extrapolation vs. interpolation) and input requirements (single-view vs. multi-view), the benchmarks and reference-based metrics (e.g., PSNR, SSIM) used in CAT3D are not applicable to our generative setting.
>
> While the CAT3D benchmarks cannot be used, we want to emphasize that our evaluation is rigorous. Our test set consists of 40 distinct scenes, spanning four different styles: indoor-real, indoor-stylized, outdoor-real, and outdoor-stylized. This is significantly more extensive than the evaluation in related single-image-to-scene works (e.g., Pano2Room evaluated on 8 real indoor scenes). This diverse benchmark validates that our method generalizes well and is not over-fitted to a particular environment.
>
> We hope this explanation clarifies why CAT3D’s protocol cannot be directly adopted for our task and demonstrates that careful consideration has been given to constructing a fair and thorough evaluation.
>
>
> **Table: Direct Comparison of Evaluation Paradigms: One2Scene vs. CAT3D (Table 1 in its original paper)**
> | Aspect | One2Scene (Ours) | CAT3D |
> |--------|------------------|-------|
> | **Primary Task** | Explorable 3D scene generation | Reconstruction |
> | **Input** | Single image | Multiple images (3/6/9) |
> | **Ground Truth** | No GT (ill-posed generation task) | Dense GT (reconstruction task) |
> | **Suitable Metrics** | No-reference metrics (NIQE, Q-Align) | Reference metrics (PSNR, SSIM, LPIPS) |
>
>
>
>
> ---
>
> **Q4: Figure 1 is confusing.**
>
> Thank you for your valuable feedback. We are sorry that the original Figure 1 may not be clear enough in its comparison with WonderJourney and DreamScene360. To address this, we have thoroughly revised Figure 1 and its caption in the main paper (please refer to our revised manuscript in the OpenReview system). The new version explicitly labels the single input image and clarifies that all other views are generated by the respective models. This new visualization more intuitively and effectively demonstrates the superiority of our method in maintaining consistency compared to baselines. We greatly appreciate the reviewer’s comment that helped us improve the clarity of this figure.

---

> ### Author Response · Authors · 2025-11-27
>
> Dear Reviewer U23k,
>
> Thank you for your precious time in reviewing our paper and your constructive comments. We appreciate if you could let us know whether our responses have addressed your concerns.
>
> Best regards,
>
> Authors of paper \#5406

---

### Official Review · Reviewer_Lk3Z · 2025-11-01

**Soundness:** 3
**Presentation:** 3
**Contribution:** 2
**Rating:** 4
**Confidence:** 3

**Summary:**

The paper One2Scene proposes a three-stage framework for generating fully explorable 3D scenes from a single image, addressing the challenges of geometric inconsistency, scale ambiguity, and semantic drift common in prior generative view synthesis methods.
The pipeline includes:
1. Panorama generation, expanding a single input image into a 360° representation using Hunyuan-Pano-DiT, providing global context.
2. Feed-forward Panorama 3D Gaussian Splatting (3DGS) network, converting the panorama into a 3D geometric scaffold by reformulating panoramic depth estimation as a multi-view stereo matching problem.
3. Scaffold-guided novel view synthesis, generating photorealistic novel views conditioned on both the 3D scaffold and anchor views through a Dual-LoRA training strategy and 3D attention fusion.
The method yields significant improvements in photorealism, geometric accuracy, and stability under large viewpoint changes, outperforming DreamScene360, WonderJourney, SEVA, and VMem across multiple metrics.

**Strengths:**

1. The overall pipeline is well-structured and intuitive, decomposing the single-image 3D generation problem into panorama expansion, geometric scaffold reconstruction, and scaffold-guided view synthesis. Each stage has clear motivation and contributes coherently to the final performance. And the paper is easy to follow with clear logistics.
2. The authors conducted comprehensive experiments with both quantitative and qualitative results.The sufficient evidence through extensive metrics makes the results solid.
3. The ablation studies are clear, for core components (Dual-LoRA Training, Memory Condition,  Bidirectional Fusion Module), there are sufficient experiments to show the effectiveness of them.
4. High efficiency and scalability. The proposed method achieves fast 3D reconstruction (0.5 s on H20, 0.1 s on H100) while maintaining visual quality, showing strong potential for real-world applications that require interactive generation like robot policy learning.

**Weaknesses:**

1. Although quantitative results are sufficient, the paper lacks explicit visualization of geometric outputs such as reconstructed point clouds or extracted scene meshes. There are many visualization with continuous frames in the website, however, geometric consistency is one of the paper’s core claims, these visualizations would provide more direct and convincing evidence of 3D structural accuracy.
2. The method has not been tested on complex, dynamic real-world environments, such as urban or outdoor settings with moving objects like pedestrians, vehicles. It remains uncertain how well the framework generalizes beyond indoor or static scenarios.
3. The pipeline depends on several large pretrained models (e.g., Hunyuan-Pano-DiT, SEVA, VGGT), which limits the degree of innovation and increases the engineering nature of the work. However, this design choice is understandable and acceptable.

**Questions:**

1. Geometric consistency validation:
Although quantitative results are comprehensive, the paper lacks explicit geometric visualizations such as reconstructed point clouds or extracted meshes. Given that geometric consistency is a central claim, could the authors provide additional evidence, to directly demonstrate 3D structural accuracy beyond continuous frame visualizations?
2. Generalization to dynamic and outdoor environments:
How does the proposed framework perform on more complex, dynamic real-world scenarios, such as urban or outdoor scenes containing moving objects (e.g., pedestrians, vehicles)? Have the authors tested or considered extensions for handling dynamic content or temporally varying geometry?

**Details Of Ethics Concerns:**

There is no ethics concern in the reviewer's opinion.

---

> ### Author Response · Authors · 2025-11-20
>
> We sincerely thank this Reviewer for the thorough review and insightful feedback. We are encouraged that this reviewer found our pipeline "well-structured and intuitive'', our experiments "comprehensive'', and our method's efficiency promising for real-world applications. We have carefully considered all comments raised by this reviewer, and we hope our following clarifications could address this reviewer's concerns.
>
> ---
>
> **Q1: Geometric Consistency Visualization.**
>
> We thank this reviewer for this constructive suggestion. We agree that direct geometric visualization is useful to substantiate our claims.
>
> To this end, we have updated our project page (https://one2scene5406.github.io) with a new section titled "To Reviewer Lk3Z: Point Cloud Reconstruction Comparison". In this section, we provide 3D point clouds, reconstructed by VGGT, from scenes generated by One2Scene and baseline methods. The visualizations provide direct and compelling evidence of our method's superior geometric precision.
> Our method produces globally consistent and structurally accurate point clouds that clearly represent the underlying scene geometry.
> In stark contrast, baselines like SEVA and VMem exhibit severe artifacts, floating fragments, and fail to form coherent 3D structures.
> This side-by-side comparison directly validates that our geometric scaffold achieves a new level of structural accuracy, which is a core contribution of our work.
>
> ---
>
> **Q2: Generalization on Outdoor Scene.**
>
> We thank this reviewer for the question. We would like to clarify that our method already demonstrates robust generalization capabilities across diverse outdoor environments, a point supported by both quantitative and qualitative evidence in our original submission.
>
> Quantitatively, as detailed in Lines 319-323 of our paper, our evaluation dataset is intentionally balanced, with outdoor scenes—including both photorealistic and artistic styles—constituting 50% of the test set. This rigorous evaluation protocol ensures that our reported metrics are not biased towards indoor scenes and can accurately reflect our method's strong performance in outdoor environments.
>
> Qualitatively, our submission's project page already includes several high-quality outdoor generations, such as "Suburban Outdoor," "Suburban Street," and "Aquatic Landscape," which serve to illustrate this versatility.
>
> To further underscore our model's capability and address this reviewer's question, we have added a new dedicated section to our project page (https://one2scene5406.github.io), titled "To Reviewer Lk3Z: Additional Outdoor Generation Results". This section contains newly generated outdoor scenes.
>
> We believe that our quantitative results and extensive qualitative examples can compellingly demonstrate that our framework is not limited to indoor environments, but is a robust and versatile solution for generating open-domain outdoor scenes.
>
> ---
>
> **Q3: Generalization on 4D Scene.**
>
> We thank this reviewer for the question. Our work is focused on generating high-fidelity static 3D scenes, which is aligned with the current state-of-the-art in the field, including prominent works such as DreamScene360, WonderJourney, and SEVA. The generation of dynamic 4D scenes is a distinct problem beyond our current scope. It is worth mentioning that even closed-source commercial entities, such as WorldLabs, are currently targeting static 3D scenes. This underscores that static scene generation remains a not well-solved challenge.
>
> While we agree that the incorporation of dynamic elements represents a compelling and vital future direction, it is a separate, dedicated research endeavor. We deliberately scoped our contribution to tackling the challenge of generating complex static 3D worlds from a single image. We posit the extension to dynamic scenes as a promising avenue for future research.

---

> ### Author Response · Authors · 2025-11-20
>
> ---
>
> **Q4: Clarification of Our Core Contributions.**
>
> We appreciate this reviewer's comment and would like to take this opportunity to clarify that our primary contribution is not the engineering integration of existing models, but rather a fundamental rethinking of the explorable 3D scene generation problem through a novel, systematic framework.
>
> Our contributions are summarized as follows:
>
> **1. A Principled Framework:** We are the first to propose a "Panorama Expansion → Geometric Scaffolding → Guided View Synthesis" pipeline. This principled decomposition systematically decouples the ill-posed problem into tractable sub-tasks, effectively addressing the critical challenges of sparse inputs, geometric ambiguity, and multi-view inconsistency that have plagued prior methods.
>
> **2. A Conceptual Shift in Geometric Estimation:** We introduce a paradigm shift by reformulating single-panorama depth estimation as a Multi-View Stereo (MVS) problem. This novel formulation allows us to leverage the powerful geometric priors learned by large-scale MVS models for robust, zero-shot panoramic 3D reconstruction. This is the key to our high-quality geometric scaffold.
>
> **3. Key Technical Innovations for Coherent Synthesis:** To realize our framework, we designed two crucial technical components: (1) A **Bidirectional Fusion Module** that enables consistent cross-view information aggregation for the MVS formulation, and (2) a **Dual-LoRA Training Strategy** that efficiently and effectively fuses the explicit geometric scaffold with the generative synthesis model, ensuring the final views are both photorealistic and geometrically faithful.
>
> These contributions, acknowledged by Reviewer 3G7Z as "well-structured", "elegant" and "intuitive" and by Reviewer 7LcG as "well designed," constitute a significant conceptual and practical advancement. Our work establishes a novel, robust, and scalable paradigm for single image based 3D scene generation. We hope the above clarification helps convey the motivations and technical significance of our work.

---

> > ### Comment · Reviewer_Lk3Z · 2025-11-26
> >
> > I appreciate the effort of the authors for preparing the rebuttal, most of my concerns are addressed. I will increase my score to 6. By the way, it would be better to inverse the coordinate of the point cloud visualization.

---

> > > ### Author Response · Authors · 2025-11-26
> > >
> > > Dear Reviewer Lk3Z,
> > >
> > > Thank you for your positive feedback. We appreciate your suggestion on the point cloud visualization and will update it accordingly in the final version.
> > >
> > > Best regards,
> > >
> > > Authors of paper #5406

---

### Official Review · Reviewer_3G7Z · 2025-11-01

**Soundness:** 3
**Presentation:** 3
**Contribution:** 3
**Rating:** 6
**Confidence:** 3

**Summary:**

One2Scene introduces a novel three-stage framework for generating geometrically consistent, explorable 3D scenes from a single image, overcoming severe distortions and artifacts that plague prior methods like DreamScene360 and WonderJourney. First, it generates a 360° panorama using Hunyuan-Pano-DiT; second, it projects this panorama into six sparse cubemap views and lifts them into an explicit 3D Gaussian Splatting scaffold via a feed-forward network that reformulates monocular depth estimation as multi-view stereo matching, leveraging robust geometric priors from large datasets through a bidirectional feature fusion module that enforces cross-view consistency; finally, it uses this scaffold as a strong geometric and appearance prior for novel view synthesis, guided by a Dual-LoRA training strategy that fuses anchor views and scaffold-rendered views via 3D-aware attention, enabling stable, photorealistic generation under large camera motions—achieving state-of-the-art results on visual fidelity, semantic consistency, and geometric stability, with a 0.5-second reconstruction time and significantly improved depth estimation accuracy over existing methods, as validated on Matterport3D and Stanford2D3D.

**Strengths:**

S1: The paper introduces a clever reformulation of monocular 3D scene generation by treating panoramic depth estimation as a multi-view stereo problem, which had not been done before.

S2: The feed-forward 3D Gaussian Splatting scaffold runs in under half a second and delivers unprecedented geometric stability without iterative refinement.

S3: The bidirectional fusion module and Dual-LoRA conditioning are elegant, practical innovations that significantly improve cross-view consistency and visual fidelity.

S4: By enabling immersive, long-range exploration from just one image—with results that surpass prior methods in both quality and robustness—it sets a new practical standard for 3D content creation.

**Weaknesses:**

W1: The method relies on a proprietary panorama generator (Hunyuan-Pano-DiT) without ablation on alternatives, limiting reproducibility.

W2: The 3D scaffold still shows minor artifacts in occluded regions under extreme rotations, suggesting room for post-processing or iterative refinement.

W3: Evaluation on stylized scenes lacks quantitative metrics for artistic fidelity—CLIP-I and NIQE may not capture stylistic coherence.

W4: No comparison to recent diffusion-based single-image 3D baselines like Bolt3D or DreamReward, which also aim for speed and realism.

**Questions:**

See weakness

---

> ### Author Response · Authors · 2025-11-20
>
> We would like to sincerely thank this reviewer for the detailed and constructive comments. We are greatly encouraged that the reviewer recognized the novelty of our multi-view reformulation, the efficiency of our geometric scaffold, and the overall practical impact of our work. We have carefully considered all the comments from this reviewer. Please find what below our itemized responses.
>
> ---
> **Q1: The method relies on a proprietary panorama generator (Hunyuan-Pano-DiT) without ablation on alternatives, limiting reproducibility.**
>
> We thank this reviewer for the question on reproducibility. We would like to clarify two key points:
>
> First, the Hunyuan-Pano-DiT generator is fully open-sourced (https://huggingface.co/tencent/HunyuanWorld-1), which makes our entire pipeline reproducible. We chose it for its state-of-the-art performance, while the alternatives we tested (e.g., DiT360 [1]) produced artifacts that would compromise our results. We have added a visual comparison on our project page: https://one2scene5406.github.io.
>
> Second, and more importantly, our framework is designed to be generator-agnostic. Our core novelty lies in the subsequent geometric and synthesis guidance, but not in the generator. This modularity is a key strength, ensuring that our contribution remains relevant and can be paired with any superior generator developed in the future.
>
> We hope that the above explanation can address this reviewer's concerns regarding the choice of generator and the reproducibility of our work.
>
> [1] Haoran Feng, Dizhe Zhang, Xiangtai Li, Bo Du, and Lu Qi. Dit360: High-fidelity panoramic image
> generation via hybrid training. arXiv preprint arXiv:2510.11712, 2025.
>
> ---
> **Q2: Minor artifacts in occluded regions under extreme rotations.**
>
> We appreciate your careful reading of our paper, and we are  encouraged that this reviewer acknowledged that our results "surpass prior methods in both quality and robustness". We concur that despite these advances, our method is not yet perfect. The minor artifacts observed in heavily occluded regions under extreme rotations are a manifestation of a fundamental challenge in single-view 3D reconstruction: the inherent ambiguity of inferring geometry that is not visible from the input viewpoint. Our method, as well as others in this domain, must make a proper guess about these unseen regions, which can sometimes lead to imperfections.
>
> This reviewer's suggestion about employing iterative optimization is highly constructive.
> An intuitive approach would be to use the newly generated view as a condition for a second round of generation. This iterative method could progressively refine the occluded regions, which would likely require the introduction of similar training data during the training process. We will explore your suggestion in our future work.
>
> ---
> **Q3: Quantitative evaluation of artistic fidelity.**
>
> We appreciate this reviewer's valuable feedback on this important aspect. While CLIP-I and NIQE are commonly used metrics in the field, they have limitations in comprehensively evaluating artistic fidelity.
>
> To address this issue, we introduce two supplementary metrics: (1) Gram matrix style loss, computed using multi-scale VGG-19 features following the seminal work in neural style transfer, which captures textural and stylistic patterns crucial for cross-view consistency; and (2) Qwen-Score, a prompt-based evaluation using the Qwen3VL-8B-Instruction model, which assesses the consistency between the generated and input views across three criteria (style, material realism, and visual fidelity), assigning a composite score from 1 (worst) to 5 (best).
>
> As demonstrated in the table below, our method achieves superior performance on both metrics, validating its efficacy in preserving artistic attributes while maintaining geometric consistency across novel views.
>
> | Methods               | Style Loss↓ | Qwen-Score↑ | NIQE↓ | Q-Align↑ | CLIP-I↑ | TransErr↓ | RotErr↓ | CamMC↓  |
> |-----------------------|-------------|-------------|-------|----------|---------|-----------|---------|---------|
> | DreamScene360         | 0.036       | 1.56        | 8.40  | 1.91     | 74.24   | -         | -       | -       |
> | WonderJourney         | 0.039       | 1.77        | 4.97  | 3.02     | 77.92   | -         | -       | -       |
> | SEVA                  | 0.009       | 3.08        | 4.53  | 3.20     | 87.82   | 0.460     | 0.165   | 0.558   |
> | SEVA + Anchor         | 0.008       | 3.57        | 4.45  | 3.45     | 88.70   | 0.422     | 0.116   | 0.460   |
> | VMem                  | 0.022       | 1.93        | 6.86  | 2.95     | 75.80   | 0.573     | 0.569   | 0.998   |
> | VMem + Anchor         | 0.016       | 2.94        | 5.23  | 3.04     | 81.33   | 0.613     | 0.426   | 0.887   |
> | **One2Scene (Ours)**  | **0.006**   | **3.88**    | **4.43**| **4.13** | **89.95**| **0.326** | **0.107**| **0.389**|

---

> ### Author Response · Authors · 2025-11-20
>
> ---
> **Q4: Comparison with Bolt3D.**
>
> We thank this reviewer for the suggestion regarding Bolt3D. While we acknowledge Bolt3D as a significant contribution to the field, which we cited and discussed in our introduction, we excluded it as a comparison baseline for two primary reasons:
>
> 1. **Fundamental Task Mismatch**: Our method is designed to generate hundreds of consistent views to construct a complete 360° scene for immersive exploration. In contrast, Bolt3D is explicitly designed for small-FoV novel view synthesis (NVS), generating only 14 sparse views. Directly applying Bolt3D to our 360° scene generation task would inevitably result in severe geometric incompleteness and large invisible regions.
>
> 2. **Reproducibility Issue**: In addition to the task mismatch, a practical barrier is that the code and pre-trained models for Bolt3D have not been publicly released. This makes a fair and rigorous quantitative comparison infeasible at this time.
>
> ---
> **Q5: Comparison with DreamReward.**
>
> DreamReward is a 3D generation model designed for **3D object generation**, whereas our work focuses on **3D scene generation**. Given this fundamental disparity in their research scopes (object vs. scene), a direct quantitative comparison is not applicable.
>
> To clarify this distinction for the reader from the outset, we revise our "Introduction" section (please refer to our revised manuscript in the OpenReview system). Specifically, we cite DreamReward in the following statement to highlight the different challenges and progress levels between object-level and scene-level generation: Although object-level 3D generation [1,2,3] has achieved rapid progress, generating an explorable 3D scene from a single image remains a formidable challenge.
>
> [1] Ruoshi Liu, Rundi Wu, Basile Van Hoorick, Pavel Tokmakov, Sergey Zakharov, and Carl Vondrick.
> Zero-1-to-3: Zero-shot one image to 3d object. In Proceedings of the IEEE/CVF international
> conference on computer vision, pp. 9298–9309, 2023.
>
> [2] Kyle Sargent, Zizhang Li, Tanmay Shah, Charles Herrmann, Hong-Xing Yu, Yunzhi Zhang, Eric Ryan
> Chan, Dmitry Lagun, Li Fei-Fei, Deqing Sun, et al. Zeronvs: Zero-shot 360-degree view synthesis
> from a single image. In Proceedings of the IEEE/CVF Conference on Computer Vision and Pattern
> Recognition, pp. 9420–9429, 2024.
>
> [3] Junliang Ye, Fangfu Liu, Qixiu Li, Zhengyi Wang, Yikai Wang, Xinzhou Wang, Yueqi Duan,
> and Jun Zhu. Dreamreward: Text-to-3d generation with human preference. arXiv preprint
> arXiv:2403.14613, 2024b.

---

> ### Author Response · Authors · 2025-11-27
>
> Dear Reviewer 3G7Z,
>
> Thank you for your precious time in reviewing our paper and your constructive comments. We appreciate if you could let us know whether our responses have addressed your concerns.
>
> Best regards,
>
> Authors of paper \#5406

---

> > ### Comment · Reviewer_3G7Z · 2025-11-27
> >
> > I thank the authors for their detailed rebuttal, which has successfully addressed most of my major concerns. Consequently, I have raised my score to 8.

---

> > ### Author Response · Authors · 2025-11-28
> >
> > Dear Reviewer 3G7Z,
> >
> > We sincerely thank you for your positive feedback and your recognition of our work!
> >
> > Best regards,
> >
> > Authors of paper #5406

---

### Author Response · Authors · 2025-11-25
**We are looking forward to hearing your further feedback**

Dear Reviewers,

Thanks for your precious time and great effort in reviewing our submission. We have uploaded our point-to-point responses to your comments. We appreciate if you could read them and let us know your further feedback.

Best regards,

Authors of paper \#5406

---

### Author Response · Authors · 2025-11-30

Dear ACs,

Following the ICLR Program Chairs' recent guidance, we are providing this summary to facilitate your assessment on our paper, \#5406. We understand the difficult circumstances and deeply appreciate your effort in getting into this process. We hope our summarization can be helpful for you to render the decision on our submission.

There are four reviews on our paper. Our rebuttal has successfully addressed the concerns from most reviewers. Consequently, three reviewers—3G7Z, Lk3Z, and 7LcG—raised their initial scores [6, 4, 6] to a strong positive consensus of [8, 6, 8]. Please kindly note that reviewers Lk3Z and 7LcG raised their scores before the data leakage incident, and reviewer 3G7Z raised his/her score from 6 to 8 during it. This indicates that the positive consensus was established based on the scientific merits of our work and our clarification in the rebuttal, which has not relation with the information leakage incident.

Another reviewer U23k did not respond our rebuttal yet due to time constraints. He/she has two major concerns. The first is the comparison with CAT3D. As we explained in the rebuttal, the code and models of CAT3D are not publicly available, and we actually compared with SEVA, which is a more advanced successor to CAT3D. The second concern is the evaluation metrics. As we explained, the protocol suggested by this reviewer is applicable to 3D reconstruction tasks, but not suitable to our task of explorable 3D scene generation.

Finally, we affirm that we have strictly adhered to all ICLR policies and principles of academic integrity throughout this process, before and after the data leakage incident. We appreciate if you could read our rebuttal and the feedback from the reviewers and consider them in your final assessment.

Thank you again for your time and your dedicated service to the community.

Sincerely,

Authors of Paper \#5406

---

### Meta-Review · Area_Chair_RuVx · 2025-12-23

**Summary:**

The paper presents One2Scene, a framework for generating explorable 3D scenes from a single image. It decomposes the problem into three stages: panorama generation, geometric scaffold construction (reformulated as a multi-view stereo problem), and scaffold-guided novel view synthesis.

The initial reception was generally positive (Ratings: 6, 6, 4, 4), with reviewers praising the clear decomposition, efficiency (0.5s reconstruction), and visual quality. Concerns were raised regarding comparisons with related methods (e.g., Pano2Room, CAT3D), geometric visualization, outdoor generalization, and reproducibility.

I recommend Acceptance. The authors provided an exemplary rebuttal that addressed all major concerns. They added direct comparisons with Pano2Room (showing superior geometric/semantic consistency), provided point cloud visualizations to prove structural accuracy, and demonstrated robust outdoor generation. Consequently, two reviewers (3G7Z, 7LcG) raised their scores to 8, and another (Lk3Z) raised to 6 . The consensus is that this work sets a new standard for single-image scene generation.

**Reviewer Concerns:**

**Addressed Concerns:**

1. Baselines (Crucial): 7LcG requested a comparison with Pano2Room. The authors added a detailed quantitative and qualitative comparison, demonstrating One2Scene's advantages in generalization (indoor/outdoor) and efficiency (0.1s vs 30 mins fine-tuning). 3G7Z and U23k asked about Bolt3D/CAT3D; the authors reasonably explained the task mismatch (NVS/Reconstruction vs. Generative Scene Extrapolation) and lack of code availability.

2. Geometric Consistency: Lk3Z requested explicit geometric visualizations. The authors added point cloud reconstructions to the project page, which convinced R2 to raise their score.

3. Generalization: Concerns about outdoor/dynamic scenes (Lk3Z, 7LcG) were addressed by pointing to the balanced test set (50% outdoor) and adding new outdoor results. The authors clarified the scope regarding static vs. dynamic scenes.

4. Reproducibility: 3G7Z's concern about the proprietary generator was addressed by clarifying that Hunyuan-Pano-DiT is open-source.

**Outstanding Concerns:**

1. Minor Artifacts: As noted by 3G7Z, there are still minor artifacts in extreme occluded regions. The authors acknowledged this as a fundamental challenge of single-view generation and proposed future directions (iterative refinement), which is acceptable.

**Reviewer Scores:**

Reviewer 3G7Z (6 -> 8): Raise to 8, stating major concerns were successfully addressed.

Reviewer 7LcG (6 -> 8): Raise to 8, satisfied with the Pano2Room comparison and practical usage discussion.

Reviewer Lk3Z (4 -> 6): Raise to 6 satisfied with the point cloud visualizations.

Reviewer U23k (4 -> likely 6): Did not update the score.

---

### Decision · Program_Chairs · 2026-01-26

Accept (Poster)